# Extreme Masking for Learning Instance and Distributed Visual Representations

Zhirong Wu[1]   Zihang Lai[2]   Xiao Sun[1]   Stephen Lin[1]

Microsoft Research Asia[1]  Carnegie Mellon University[2]

Reviewed on OpenReview: `https://openreview.net/forum?id=3epEbhdgbv`

## Abstract

The paper presents a scalable approach for learning spatially distributed visual representations over individual tokens and a holistic instance representation simultaneously. We use self-attention blocks to represent spatially distributed tokens, followed by cross-attention blocks to aggregate the holistic image instance. The core of the approach is the use of extremely large token masking (75%-90%) as the data augmentation for supervision. Our model, named ExtreMA, follows the plain BYOL approach where the instance representation from the unmasked subset is trained to predict that from the intact input. Instead of encouraging invariance across inputs, the model is required to capture informative variations in an image.

The paper makes three contributions: 1) It presents random masking as a strong and computationally efficient data augmentation for siamese representation learning. 2) With multiple sampling per instance, extreme masking greatly speeds up learning and improves performance with more data. 3) ExtreMA obtains stronger linear probing performance than masked modeling methods, and better transfer performance than prior contrastive models.

## 1 Introduction

Masked modeling (Devlin et al., 2019) has emerged as a viable approach for visual representation learning. On a generic transformer architecture (Vaswani et al., 2017), it optimizes a learning objective based on the masked signal prediction popularized in natural language understanding (Devlin et al., 2019), without reliance on heavily engineered image augmentations (Wu et al., 2018; Chen et al., 2020b; Grill et al., 2020). Superior finetuning performance has been demonstrated with this approach; however, the pretrained representation does not work competitively off-the-shelf (Bao et al., 2021), e.g., for k-nearest-neighbor retrieval.

On the other hand, Siamese networks trained with contrastive objectives (Oord et al., 2018) are strong for learning off-the-shelf representations (Radford et al., 2021). This fundamental difference lies in the way they represent data. Siamese networks extract a holistic instance representation for an image, whereas masked modeling acquires a spatially distributed representation over individual tokens that comprise an image [1]. No instance representation is explicitly modeled or provided supervision in masked modeling approaches (Devlin et al., 2019; Bao et al., 2021).

In this paper, we study the connections between the instance and the distributed representations, and we explore self-supervision for learning these representations. We start with an observation that random masking could be viewed as a novel data augmentation scheme not previously exploited in Siamese networks. For the masked area, its potential degrees of freedom grow combinatorially large with its size, allowing for richer self-supervision than conventional augmentations such as cropping and scaling, which are heavily

---

[1]In this paper, the term instance representation refers to the feature representation for the holistic image, not for object instances used in instance segmentation literatures. The term distributed representation intends to mean that the image is represented by a pattern of activity distributed over patch tokens.

biased towards areas around the image center (He et al., 2022). More importantly, the self-supervision from conventional augmentations lead to a common representation vector that encompasses multiple augmentations of an instance (Tian et al., 2020; Purushwalkam & Gupta, 2020), and this invariance may degrade the sensitivity of the representation to spatial locality. On the contrary, random masking preserves the original content of the unmasked area and the geometrical structure of the data.

We propose a simple model, called ExtreMA, where the instance representation from masked input is trained to predict that from the full view, in a plain BYOL fashion (Grill et al., 2020). The information gap created by masking encourages the student network to encode as much information as possible, and hence bootstrap the teacher network to be stronger. We adopt the vision transformer ViT (Dosovitskiy et al., 2021) to embed distributed representations over patches, and this is followed by cross-attention blocks (Touvron et al., 2021b) to aggregate the distributed representations into the instance representation. The instance-level learning objective provides the supervision for both of the representations. In our model, the distributed representations are only implicitly learned without the corresponding token-level objective used in masked modeling. However, through investigating and visualizing the attention maps (in Figure 1), we find that the output representations of our model maintain accurate correspondences with the input tokens and that semantic clusters tend to emerge from the learned distributed representations.

A notable distinction of this model is its effectiveness with an extremely large masking ratio (75% - 90%), while typical masked modeling approaches work best between the range of 50% to 75% (El-Nouby et al., 2021; Bao et al., 2021; He et al., 2022). Besides the computational efficiency that this brings, a key aspect of extreme masking is the complementarity that arises among multiple samples. Due to the high redundancy in visual data, the visible content for samples with different masking becomes independent only when the masking ratio becomes large. In addition, multiple masks speed up learning and convergence significantly, making the system a fast learner that is hungry for more data. In practice, multiple sampling is also computationally appealing, as the teacher network for processing the full content needs only to be forwarded once.

ExtreMA enjoys the favorable properties of Siamese representation learning. The instance representation from the model can be used off-the-shelf for measuring semantic similarities. The framework also welcomes other data augmentations besides masking for applications with different ends. However, unlike conventional contrastive learning, ExtreMA does not rely on data augmentation induced invariances to achieve generalization. Rather, ExtreMA preserves all possible useful information from the masked view in order to recover the full image. The generative aspects of our model are exhibited in Figure 2. It can faithfully inpaint the masked pixels through network inversion (Zhao et al., 2020). Moreover, the instance representation is shown to be sensitive to spatial and scale variations for localizing objects in Figure 3. These properties demonstrates that ExtreMA learns both instance and distributed representations that well captures scale, location, and color intensities.

In the experiments, we systematically study the model behavior under different masking ratios, its convergence properties using multiple masks on larger datasets, and integration with various other data augmentations. Based on the study observations, we also propose a new augmentation scheme which uses shared image crops but different colors for the two input views. Our main results on ImageNet1k outperform prior masked modeling approaches on both finetuning and linear probing metrics. Notably, this is achieved by training ExtreMA using a single node of 8×V100 GPUs in about two days for a ViT-Base model. We also evaluate the transfer performance for semi-supervised learning and semantic segmentation. For both applications, ExtreMA produces superior results compared with prior arts.

## 2 Related Works

In self-supervised representation learning, labels are mined from the data itself to achieve generalization beyond that from human annotations, especially when the training data is at scale. Past works demonstrate generalization through k-nearest-neighbors (Wu et al., 2018) and zero-shot classification on the learned features (Radford et al., 2021), or finetuning the model for a limited schedule (Chen et al., 2020c; He et al., 2022). The central problem under investigation is how to extract the training labels automatically and formulate the pretext tasks. In high-level vision, such pretext tasks include predicting colors from a grayscale image (Zhang et al., 2016), inpainting pixels given the spatial context (Doersch et al., 2015) or through

autoregression (Chen et al., 2020a), predicting the orientation of a rotated image (Komodakis & Gidaris, 2018), solving a jigsaw puzzle given shuffled patches (Noroozi & Favaro, 2016), and others (Donahue & Simonyan, 2019; Zhang et al., 2017). The key idea is that the network has to learn semantics in order to solve the pretext tasks. Recently, there has been a resurgence of the context prediction pretext task (Bao et al., 2021; Li et al., 2021) that has accompanied the rise of vision transformers (Dosovitskiy et al., 2021). Input tokens are masked, and the model is trained to predict the masked tokens from the visible tokens in a BERT fashion. The target tokens could be represented by dVAE tokens (Bao et al., 2021), raw pixel values (He et al., 2022), or features (Wei et al., 2022; Dong et al., 2021) from an online learned encoder (Zhou et al., 2021; El-Nouby et al., 2021; Baevski et al., 2022; Chen et al., 2022; Tao et al., 2022). Such representations are shown to surpass prior art when finetuned for downstream tasks (He et al., 2022).

Contrastive learning is a special pretext task of instance discrimination to learn view-invariant representations from data augmentations. It encourages different views of an image instance to have similar representations relative to negative samples (Wu et al., 2018; He et al., 2020; Chen et al., 2020a), negative clusters (Caron et al., 2020), or even without using negatives (Grill et al., 2020) at all. Views of an image are commonly processed by a Siamese network (Chen et al., 2020b; Li et al., 2022) with a momentum encoder (He et al., 2020) on one of its branches. To boost performance, the community has crafted various data augmentations including color jittering (Wu et al., 2018), Gaussian blurring (Chen et al., 2020b), solarization (Grill et al., 2020), and copy-and-paste (Zhao et al., 2021), as well as determined their optimal hyper-parameters. Multiple augmentations (Fort et al., 2021) per input are investigated to improve test time performance and convergence speed for supervised learning. In self-supervised learning, DINO uses 8 local crops to improve representation quality. Its effect on convergence speed remains under explored. Contrastive models trained at scale are shown to perform on par with supervised learning (Goyal et al., 2019; 2021). The contrastive learning framework has the flexibility to handle various data augmentations, while traditional pretext tasks need non-trivial engineering to be trained in a multi-task manner (Doersch & Zisserman, 2017). Recently, there have been efforts (El-Nouby et al., 2021; Zhou et al., 2021; Mishra et al., 2022) to combine contrastive learning with masked modeling objectives in a multi-task manner. The two tasks complement each other, but the intrinsic connection remains unclear.

The technique of masking originates from representation learning on languages (Devlin et al., 2019; Liu et al., 2019) and is especially suited for transformer architectures. In computer vision, block-wise masking (Bao et al., 2021) and random masking (He et al., 2022) are investigated to cope with the 2D nature of images. Aside from BERT-like training approaches (Bao et al., 2021; He et al., 2022; El-Nouby et al., 2021; Zhou et al., 2021; Baevski et al., 2022), a special type of random masking in the form of small local image crops has also been adopted in contrastive models (Caron et al., 2021; 2020). However, the local crop augmentation introduced in (Caron et al., 2020) is mainly designed for computational efficiency, without revealing the impact of substantial content removal on representation quality. A recent work MSN (Assran et al., 2022) explores the application of masking in Siamese networks. However, its masking ratio is low (30% for ViT-base), and it heavily relies on other augmentations besides masking to work. ADIOS (Shi et al., 2022) adversarially optimizes masking as a novel data augmentation for contrastive learning. It shows masking is complementary to existing conventional image augmentations. Data2vec (Baevski et al., 2022) formulates a masked image prediction task, with the prediction target bootstrapped from a momentum encoder. Similar to Data2vec, our work explores representation learning with the Siamese architecture and the masking mechanism. Differently, our model predicts the holistic instance representation without using the mask token throughout the network. Hybrid models (Zhou et al., 2021; Tao et al., 2022; Mishra et al., 2022) of contrastive learning and masked image modeling formulate a multi-task problem to enjoy the best of two paradigms. However, the technique of masking is mainly to supervise masked modeling, not as an augmentation for contrastive learning. Specifically, Zhou et al. (2021) finds that adding masking augmentation actually hurts representation quality. Our work reveals the power of masking even when used as a data augmentation, without explicit supervision for token-level predictions.

## 3 The ExtreMA Approach

This work explores the use of random masking as data augmentation for Siamese representation learning with instance-level supervision. The model uses an architecture of two parallel networks, where the momentum

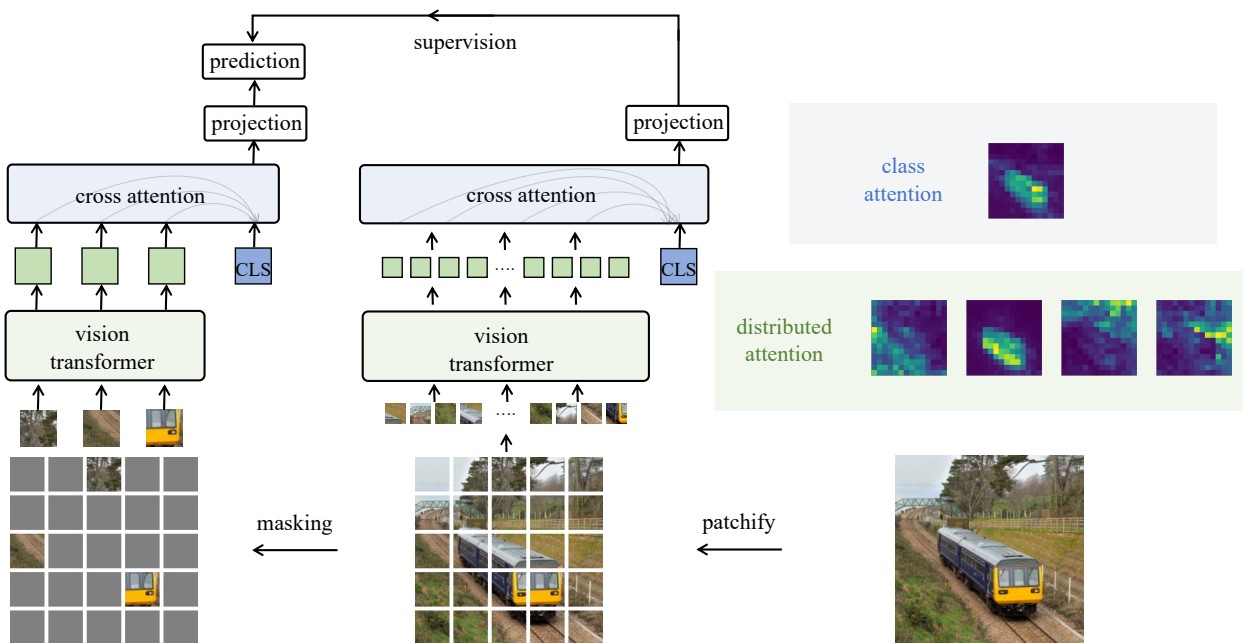

Figure 1: **Overview of ExtreMA.** Our model follows the Siamese network approach for representation learning. The momentum encoder processes the full view while the base encoder processes a partial view from extreme masked sampling. Input tokens are encoded into distributed representations via vision transformers and gathered into an instance representation via cross-attention blocks with an appended $[CLS]$ token. Self-supervision is applied at the instance level. We visualize attention maps for four query patches in the last layer of ViT and cross attention for the class token.

encoder processes the full image crop and the base encoder processes the masked image crop. The information gap, i.e. the masked image region, is the basis of the supervision for training the base encoder. An overview of our approach is illustrated in Figure 1. We describe details about masking and the model architecture as follows.

**Extreme Masking.** Given an image, we first divide it into non-overlapping patches to be fed into the vision transformer. A fixed sinusoidal positional encoding is added to each embedded patch, and a few of the embedded patches are sampled randomly (He et al., 2022) according to the masking ratio.

A key aspect of our approach is that it achieves its best performance with an extremely large masking ratio of 75%-90%, leaving the base encoder to process just a fraction (10%-25%) of the patches. This is in contrast to masked image modeling where the performance degrades when the masking ratio exceeds 75% (He et al., 2022). The extremely high masking ratio sets a very hard pretext task for the network. We hypothesize that the ability of ExtreMA to succeed with extreme masking is in part due to the momentum encoder processing the entire image, whereas the encoder in masked image modeling never receives the full view as input. This may create train-test discrepancy for masked modeling, as the attention blocks need to generalize from processing a fraction of the tokens to the full set.

Extreme masking can be further assisted with the multi-masking technique, where multiple paralleled base encoders processing different masked versions of the image are used to accelerate learning. Multiple student networks can share the same learning target from the momentum encoder, which makes multi-masking an efficient learner. We note that such a multi-masking technique is not immediately applicable or efficient for masked modeling since the decoder needs to run independently for each masked input without being able to share the learning target.

Extreme masking also introduces new challenges. We observe that extreme masking tends to overfit to the training data especially when multi-masking is enabled. Concretely, the model obtains good classification performance on the training dataset, but less generalizable to unseen data. This is different from "cheating",

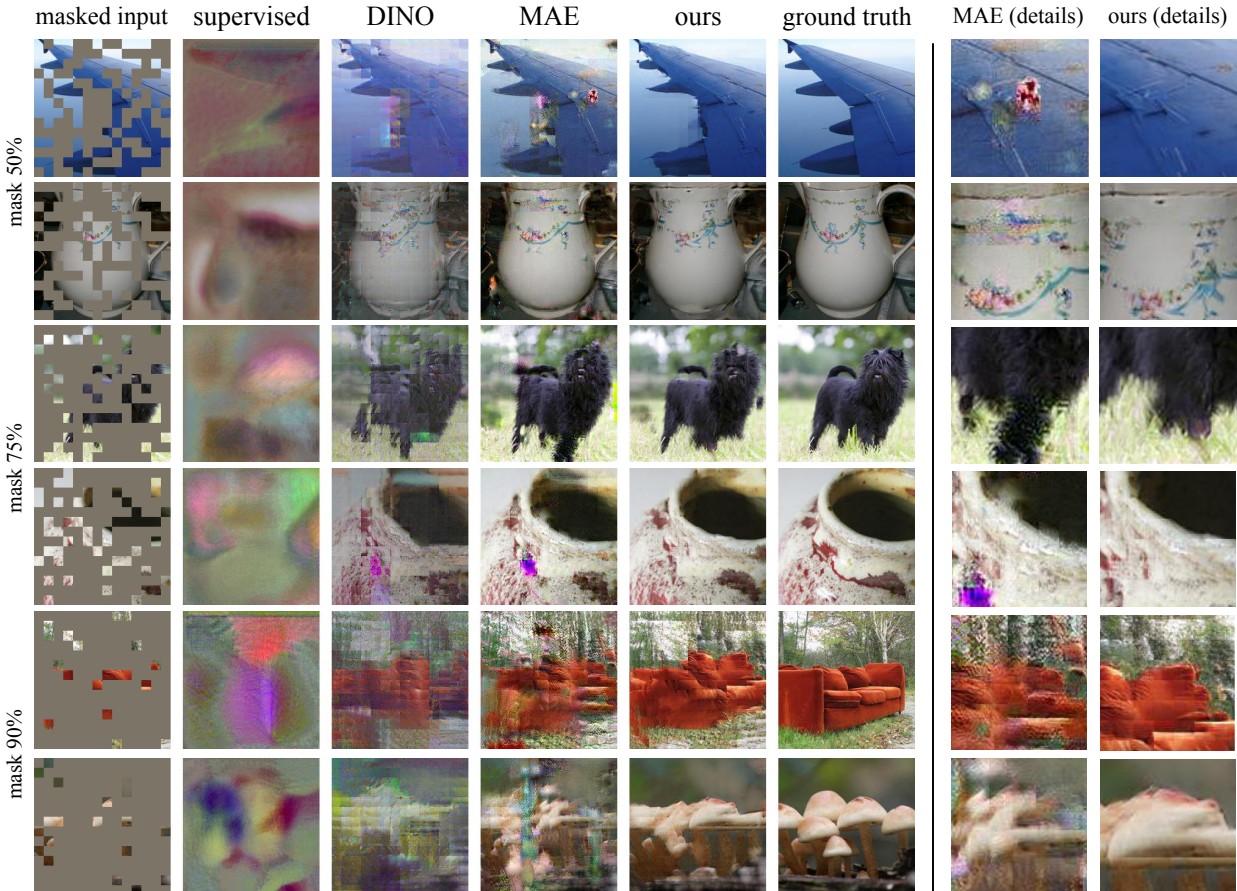

Figure 2: **Generative properties of the distributed representations at various masking ratios.** We use the deep image prior technique to invert the representations. Our reconstruction result shows the best quality overall. Supervised ViT fails to produce meaningful content. MAE fails to inpaint proper colors due to the use of normalized pixels. DINO loses information about spatial locality.

as the feature effectively learns semantics on the training data from self-supervision. Using more unlabeled training data is a viable way to mitigate the overfitting phenomenon. This is discussed further in the experiment section.

**Distributed and Instance Representations.** We adopt the vision transformer (Dosovitskiy et al., 2021) for its efficiency and flexibility in handling input content of variable size. The vision transformer embeds the input visual tokens into a spatially distributed representation via the self-attention mechanism. An instance representation is desired in order to allow supervision from the instance level. To achieve this, we use cross-attention blocks (Touvron et al., 2021b) to aggregate the distributed patch-level representations into a single representation with an additional appended class token. Only the class token acts the query in the cross-attention blocks while the patch tokens remain frozen without updates. This makes the cross-attention blocks lightweight with O(N) complexity compared to O($N^2$) in self-attention. The overall encoder architecture follows CaiT-style (Touvron et al., 2021b) design. The projection head and the prediction head follow the instance representation.

We have investigated two other alternatives to represent an instance, but neither works well. If we feed the instance token as an input to the transformer as in ViT, optimization becomes unstable, potentially due to the large masking rate for the input. With average pooling over the token representations as the instance representation, the model finds a shortcut on learning averaged patch features without learning attention across patches. Details are provided in the appendix.

**Learning Objective.** The instance representation from the masked input is trained to predict that from the unmasked input by simply minimizing the cosine distance between the two representations. We primarily follow BYOL for simplicity in this paper, but our approach is also found to work with the contrastive loss (Wu et al., 2018; Oord et al., 2018) in our experiments. Our learning objective differs with the conventional BYOL in the following aspect. Conventional BYOL adopts a symmetric loss where the two views are learned to predict each other. This will drive the representation to find the common subspace shared by the two views, with invariance over other information (Tian et al., 2020). In our case, since the shared information between the two views is obvious, finding the commonality between the views does not make for meaningful self-supervision. Thus, we adopt the asymmetric loss where only the masked view is trained to predict the intact view. The information gap created by masking encourages the network to extrapolate the masked regions, instead of seeking a shared subspace.

Compared with masked modeling approaches, our learning objective is fundamentally different. The instance level supervision does not explicitly enforce spatial reasoning for each individual token. Nonetheless, we find strong evidence that our model learns a distributed representation over the tokens. We visualize the attention maps for four query patches in the last layer of the transformer block in Figure 1. The shown visualization is averaged across 12 attention heads. We observe that patch tokens tend to group into meaningful semantic clusters.

**BYOL Details.** Our design choices for the projection and the prediction head are even simpler than the original BYOL (Grill et al., 2020). We replace the BatchNorm with LayerNorm and the ReLU activations with GeLU activations, making the overall framework free of BatchNorm and consistent with the rest of the transformer blocks. Our work also incidentally demonstrates that BYOL does not rely on BatchNorm to prevent collapse (Richemond et al., 2020). The projection head and the prediction head have 3 and 2 hidden layers respectively, a hidden dimension of 4096, and an output dimension of 256, following the original design.

**Limitations.** It remains as a limitation without being able to fully understand why CaiT-style architecture (Touvron et al., 2021b) is crucial for stable convergence and why conventional ViT architecture fails. More discussions to this problem is included in the appendix. When training the ExtreMA model on a dataset, the training epochs may need to be validated and tuned in order to prevent overfitting. The optimal masking ratio may also varies depending on the property of the dataset.

## 4 Representation Properties

To understand the distributed representation and the instance representation trained with extreme masking augmentation, we invert the distributed representations into the pixel space and examine the sensitivity to locality of the instance representation. These results give further evidence that the model learns a meaningful distributed representation without a BERT-like objective, and that the instance representation preserves detailed visual information. The following analysis is not meant to compare different models in a quantitative way, but to provide intuitive insights on what have been learned by the models.

**Generative Properties.** Given a pretrained model and a masked image, we can encode the visible patches using the distributed representations, and invert these partial representations back to the pixel space to reconstruct the masked patches. Specifically, we follow the technique of deep image prior (Ulyanov et al., 2018) and minimize the L2 distance on the visible representations between the masked image and the reconstruction. Details of the method is included in the appendix. This reconstruction technique allows us to examine the content of the encoded features without the need to further train a new generative model. Figure 2 shows the reconstruction result. We vary the masking ratio and compare results with supervised DeiT (Touvron et al., 2021a), DINO and MAE [2] using this inversion method. Supervised DeiT is unable to produce any meaningful content. MAE fails to inpaint proper colors due to the use of normalized pixels. DINO is inaccurate with spatial localities. Our result is spatially smooth and accurate in color. The inversion technique suffers when the masking ratio is very large, due to limited ability to inpaint unseen semantic areas.

---

[2]We acknowledge that MAE is able to do amazing reconstruction using its decoder. The inversion experiment for MAE is mainly to investigate its encoder in a fair comparison with competing approaches.

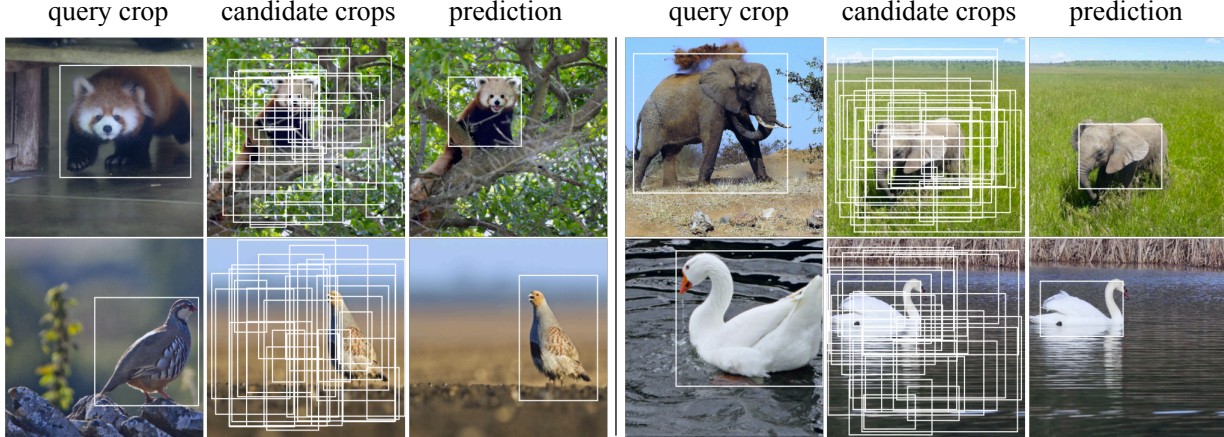

Figure 3: **ExtreMA is sensitive to spatial and scale variations.** We randomly sample 25 candidate bounding boxes of 5 scales and 5 random locations from a test image and we use the query crop to retrieve the closest bounding box in the test image. The highest ranked crop is shown as the prediction. The instance representation from our model is able to identify the correct scale and location, suggesting that ExtreMA is sensitive to information beyond semantics.

Table 1: Mask ratio.

| ratio | ft. | lin. |
|-------|------|------|
| 50% | 81.9 | 36.3 |
| 70% | 82.3 | 64.4 |
| 80% | 82.4 | 67.3 |
| 85% | 82.4 | 66.3 |
| 90% | 82.3 | 61.6 |
| 95% | 81.6 | 49.3 |

Table 2: Multi-masks trained on IM1k.

| ratio 75% | | | ratio 80% | | | ratio 90% | | |
|-----|------|------|-----|------|------|-----|------|------|
| num | ft. | lin. | num | ft. | lin. | num | ft. | lin. |
| 1 | 82.4 | 64.8 | 1 | 82.4 | 67.3 | 1 | 82.3 | 61.6 |
| 2 | 82.7 | 67.2 | 2 | 82.6 | 68.8 | 2 | 82.5 | 64.0 |
| 4 | 82.9 | 67.7 | 4 | 82.8 | 67.5 | 4 | 82.7 | 63.1 |
| | | | 5 | 82.9 | 67.1 | 8 | 82.9 | 60.3 |
| | | | | | | 10 | 83.0 | 59.3 |

Table 3: Trained on IM22k.

| ratio 90% | | |
|-----|------|------|
| num | ft. | lin. |
| 1 | 82.5 | 60.4 |
| 2 | 82.7 | 65.6 |
| 4 | 83.0 | 69.0 |
| 8 | 83.2 | 71.4 |
| 10 | 83.2 | 72.0 |

**Locality Properties.** We use the k-nearest neighbor technique to probe the instance representation. We first generate a small gallery set by random sampling of image crops that vary spatially and in scale from a single image. We then use another query image crop from the same semantic category to rank the gallery set. We resize these image crops to 224 × 224 and extract the instance representations for measuring similarities. In Figure 3, the top nearest retrieval returns the image bounding box with the closest spatial and scale configuration as the query crop. The results suggest that the instance representation is sensitive to spatial and scale changes, and the learned high-level representation is not a result of invariance but is more powerful and generalizable. This example also demonstrates a form of zero-shot detection using exemplars (Malisiewicz et al., 2011).

## 5 Experiments

### 5.1 Ablation Studies

We pretrain the representation on ImageNet and evaluate it on finetuning (ft) and linear probe (lin) in our ablations. We finetune the model on top of the distributed representation, and conduct linear probes with the instance representation. The evaluation protocol mainly follows BEiT and MAE.

**Implementation Details**. We use the original ViT-base (Dosovitskiy et al., 2021) as the backbone architecture without the layer scale technique (Touvron et al., 2021b). The class attention follows the original design in (Touvron et al., 2021b) with a default of two transformer blocks and a layer scale hyper-parameter of 0.1. We train our model using the AdamW optimizer (Loshchilov & Hutter, 2018) with a batch size of 2048, an initial base learning rate of 1.5e-4, and a weight decay of 0.1. The exponential averaging weight for

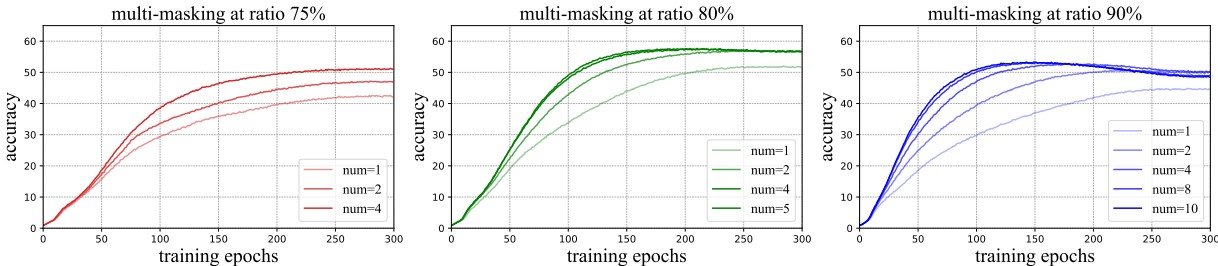

Figure 4: Convergence curves for multi-masking on ImageNet1k. Each plot shows kNN accuracy on the validation set with respect to training epochs. Multi-masking with 80% and 90% ratio enjoys steep learning curves, but suffers from overfitting on ImageNet1k.

Table 4: Other augmentations.

| augment | 1 aug | | 2 augs | |
|---|---|---|---|---|
| | ft. | lin. | ft. | lin. |
| none | 82.3 | 55.1 | - | - |
| color | 82.5 | 62.0 | 83.3 | 62.4 |
| rand size crop | 82.6 | 68.8 | 82.2 | 66.8 |
| crop + color | 82.6 | 69.0 | 83.1 | 73.3 |
| shared crop + color | - | - | 83.3 | 73.1 |

Table 5: Cross attention block.

| #blocks | ft. | lin. |
|---|---|---|
| 1 | 82.6 | 66.2 |
| 2 | 82.6 | 68.8 |
| 3 | 82.6 | 68.5 |

Table 6: Training loss.

| objective | ft. | lin. |
|---|---|---|
| BYOL | 82.6 | 68.8 |
| InfoNCE | 82.8 | 66.8 |

the momentum encoder is initialized to 0.996 and increased to 1.0 following a cosine schedule. The default augmentation is random resized cropping and random flipping. All models are trained for 300 epochs.

**Masking Ratio.** We first vary the masking ratio using a single mask for training. In Table 1, the finetuning performance plateaus across a wide range from 70% to 90%, while the linear probe performance peaks at ratio 80%. Notably, an extremely large masking ratio of 90% also achieves reasonably good performance. The performance degrades beyond 90%.

**Multi-Masking and Convergence Speed.** For each image instance, we generate multiple masks without replacement for the student network. The loss as well as the gradient are averaged over multiple masked inputs for each parameter update. The learning rate schedule is unaffected by multi-masking and kept unchanged with single masking. We investigate the behavior of multi-masking under the ratios of 75%, 80%, and 90% in Table 2. Finetuning performance consistently improves with more masked inputs. However, the linear probe performance degrades when too many masks are used, especially when the masking ratio gets larger. We take a close look at this phenomenon and find that the training accuracy for linear probing actually improves with greater multi-masking. This suggests that the model overfits to the training data without using labels. In Figure 4, we plot the k-nearest-neighbor classification curves on the validation set with respect to training epochs. The hyper-parameter $k$ is set to 200 and the gallery is set to 10% of the ImageNet training set. Masking with ratio 75% does not suffer from overfitting with multi-masking, but converges less quickly. Multi-masking with extreme ratio 90% has the steepest learning curve, but it tends to saturate and degrade after 120 epochs. We hypothesize that this is because masked inputs become more independent and bring complementary learning signal when the masking ratio grows larger.

To combat overfitting while preserving fast learning, a straightforward solution is to use larger datasets. We therefore study multi-masking on ImageNet22k, which is about 10 times larger in total images. We train the model for 30 epochs, which maintains the effective number of optimization iterations and reveals the impact solely from data scale. The evaluations for finetuning and linear probing are all conducted with ImageNet1k. In Table 3, at the masking ratio of 90%, ExtreMA no longer suffers from overfitting as the number of masks increases. The model performance is also consistently better than using ImageNet1k training data. This shows that our model benefits from more data for large-scale representation learning.

**Other Augmentations.** Supervision at the instance level enables integration of other augmentations for both the student and teacher networks. We set the default masking ratio to 80% with two masks in the following ablations, as it does not suffer from overfitting. We consider the augmentations of cropping (random

Table 7: ImageNet1k classification comparison.

| methods | epochs | ViT-S | | ViT-B | |
|---|---|---|---|---|---|
| | | ft. | lin. | ft. | lin. |
| MoCo-v3 | 300 | 81.5 | 73.1 | 83.2 | 76.2 |
| DINO | 400 | 81.7 | 77.0 | 83.6 | 78.2 |
| MSN | 600 | 81.6 | 76.9 | 83.4 | 76.8 |
| BEiT | 800 | - | - | 83.2 | 37.6 |
| MAE | 1600 | - | - | 83.6 | 67.8 |
| Data2vec | 800 | - | - | 84.2 | 60.8 |
| ExtreMA (1k) | 300 | 81.8 | 69.4 | 83.7 | 73.3 |
| ExtreMA (22k) | 30 | 81.5 | 65.7 | 83.9 | 74.5 |

Table 8: ViT-B Wall-clock time comparison using a single node of 8×V100 GPUs.

| methods | epochs | time |
|---|---|---|
| DINO | 400 | 300 hrs |
| MSN | 600 | 700 hrs |
| BEiT | 800 | 240 hrs |
| MAE | 1600 | 650 hrs |
| Data2vec | 800 | 250 hrs |
| ExtreMA (80% ratio ×1) | 300 | 29 hrs |
| ExtreMA (80% ratio ×2) | 300 | 36 hrs |
| ExtreMA (80% ratio ×5) | 300 | 60 hrs |

resized crop + flipping) and color (color jittering and random grayscaling). We do not consider Gaussian blurring and solarization as their effects are marginal.

We first examine the case of a single augmentation, where the input of the teacher branch is augmented and the student branch takes a random masking sample of the teacher's input. Such a scheme is akin to enlarging the training dataset without introducing other supervision from augmentations. As reported in Table 4, by just using a center crop, our model achieves a reasonable result of 82.3% finetuning and 55.1% linear probing performance. Color and cropping augmentations improve the overall performance individually but their effects are marginal when both are used.

We next consider two independent augmentations, one for the student and one for the teacher, with the student's input undergoing masking as well. The self-supervision in such a scheme introduces invariance, such as spatial, scale, and color intensity, similar to prior contrastive models (Caron et al., 2021; Chen et al., 2021). Crucially, we find that adding spatial and scale invariance by two crops of an image may hurt representation quality, with finetuning decreased by 0.4% and linear probing decreased by 2.0%. On the other hand, color invariance is shown to be beneficial, leading to a significant 0.8% gain from 82.5% to 83.3%. Using cropping and color augmentations combined, the linear probing performance improves substantially to 73.3% while the finetuning performance drops 0.2% as spatial invariance may hurt generalization.

Based on these observations, we propose another augmentation scheme that uses a shared spatial crop for the two network branches, but two different color augmentations. Such scheme achieves the best overall performance for finetuning and linear probing.

**Cross Attention Blocks.** We use cross-attention heads (Touvron et al., 2021b) to aggregate the distributed representations into the instance representation. We ablate the number of blocks for this design in Table 5. The finetuning performance is not affected by the depth of the cross-attention blocks, while the linear probing performance is improved by 2% with two blocks and saturates for more blocks.

**Training Objective.** Besides BYOL, ExtreMA also works with other Siamese representation learning objectives, such as InfoNCE (Oord et al., 2018) with negatives. In Table 6, we provide the result with a MoCo-v3 implementation using a contrast temperature of 0.2. Compared with the BYOL objective, the finetuning performance is improved by 0.2% and linear probing drops by 2.0%.

## 5.2 ImageNet Comparisons with Previous Methods

We compare with representative contrastive methods MoCo-v3, DINO and MSN, as well as masked image modeling methods BeiT, MAE and Data2vec on ImageNet classification. We use our strongest model with five masks of ratio 80% and color augmentations. The finetuning takes 200 epochs for ViT-S and 100 epochs for ViT-B following prior works. The results [3] are summarized in Table 7. Our approach outperforms MoCo-v3, DINO, MSN, MAE, BEiT for finetuning evaluations but underperforms Data2vec. This is likely due to the layer-averaged targets in Datavec provides substantial improvement. Our linear probing outperforms the masked image modeling methods by a large margin but underperforms contrastive counterparts. This may

---

[3]We note that ExtreMA can be reproduced stably with small variance of 0.1% for linear evaluation and finetuning.

Table 9: Semantic segmentation on ADE20K.

| methods | epochs | mIoU |
|---|---|---|
| DINO | 400 | 47.2 |
| MSN | 600 | 47.1 |
| MoCo-v3 | 300 | 47.3 |
| BEiT | 800 | 47.1 |
| MAE | 1600 | **48.1** |
| ExtreMA (1k) | 300 | 47.9 |
| ExtreMA (22k) | 30 | **48.4** |

Table 10: Semi-supervised classification.

| methods | epochs | 1% | 10% |
|---|---|---|---|
| scratch | - | 9.0 | 44.8 |
| BEiT | 800 | 35.9 | 69.7 |
| DINO | 400 | 64.7 | 75.9 |
| MoCo-v3 | 300 | 57.2 | 75.8 |
| MSN | 600 | 66.6 | **76.8** |
| MAE | 1600 | 52.7 | 72.1 |
| ExtreMA (1k) | 300 | **67.3** | 76.1 |

be due to the lack of global crops and other heavy image augmentations. Since no holistic representation is modeled for the image in masked modeling, BEiT, MAE and Data2vec perform less competitive for linear probing with features average pooled from tokens. The ViT-S model does not scale well with large data, potentially limited by its model size.

A notable advantage for ExtreMA is its computational efficiency and fast convergence speed. This allows us to train ViT-Base models of 300 epochs using a single node of 8×V100 GPUs for 29 hours to 60 hours depending on the choice of multi-masking. Such hardware requirement is friendly to resource-limited academic labs. On the contrary, prior self-supervised representation models require multi-node training and lengthy optimization. We summarize the wall clock times of representative models in Table 8. Since official releases of prior methods are reported with multi-node training, we estimate their wall-clock time using just a single node of 8×V100 GPUs. ExtreMA achieves 5× to 10× speedups for visual representation learning.

## 5.3 Transfer Learning Results

We consider two transfer learning scenarios with limited target labels: semi-supervised image classification and semantic segmentation. For both experiments, we use our strongest model with five masks of ratio 80% and color augmentations.

**Semi-supervised Learning.** Given the pretrained model, we use a small fraction of the ImageNet1k training labels (1% or 10%) for semi-supervised finetuning. We append the classification head on the first output layer of the projection head following SimCLR-v2 Chen et al. (2020c). The finetuning protocol and data augmentation mainly follows BEiT. The model is optimized using AdamW with an initial learning rate of 5e-6 for 1000 epochs and a batch size of 1024. Comparison results are shown in Table 10. ExtreMA outperforms the masked image modeling approaches MAE and BEiT by a large margin of 12% and 30% using 1% of the labels. Surprisingly, ExtreMA obtains better results than DINO, which is heavily tuned for ImageNet classification with higher linear probing performance than our approach. ExtreMA is also on par with MSN which is specifically designed for semi-supervised learning. The models pretrained with ImageNet22k perform worse, because the majority of unlabeled classes are less relevant to the target 1k classes. We thus follow the prior evaluation practice and omit the ImageNet22k entry.

**Semantic Segmentation.** We evaluate semantic segmentation performance on the ADE20K (Zhou et al., 2017) dataset. Following prior works, we initialize the UperNet framework (Xiao et al., 2018) using our pretrained model and finetune the segmentation model end-to-end. The model is optimized using AdamW for 80k iterations with an initial learning rate of 1e-4 and a batch size of 16. We set the weight decay to 0.05 and layer-wise learning rate decay to 0.85. The results are shown in Table 9. Our method is able to outperform competitive representation learning baselines such as DINO, MSN, MoCo-v3 and BEiT. Our model is even comparable to MAE, which is trained with a much heavier schedule (300 epochs vs. 1600 epochs). By scaling the training data to the larger ImageNet-22K dataset while keeping the total number of iterations unchanged, our model performance improves by 0.5 mIoU, surpassing all prior arts by a significant margin. This indicates that our model scales well with data.

**Object Detection and Instance Segmentation.** We evaluate the transfer performance on the MSCOCO dataset. We adopt the Mask-RCNN framework for object detection and instance segmentation using the

Table 11: Object detection and instance segmentation transfer on COCO.

| methods | epochs | object detection | | | instance segmentation | | |
|---|---|---|---|---|---|---|---|
| | | $AP$ | $AP_{50}$ | $AP_{75}$ | $AP$ | $AP_{50}$ | $AP_{75}$ |
| DINO | 400 | 46.8 | 68.6 | 50.9 | 41.5 | 65.3 | 44.5 |
| MSN | 600 | 45.8 | 68.2 | 49.5 | 40.6 | 64.8 | 43.2 |
| MoCo-v3 | 300 | 45.5 | 67.1 | 49.4 | 40.5 | 63.7 | 43.4 |
| BEiT | 800 | 42.1 | 63.3 | 46.0 | 37.8 | 60.1 | 40.6 |
| MAE | 300 | 45.4 | 66.4 | 49.6 | 40.6 | 63.4 | 43.7 |
| MAE | 1600 | 48.4 | 69.4 | 53.1 | 42.6 | 66.1 | 45.9 |
| ExtreMA (1k) | 300 | 47.5 | 68.9 | 51.9 | 42.0 | 65.6 | 45.1 |
| ExtreMA (22k) | 30 | **48.5** | **69.8** | **53.1** | **42.7** | **66.5** | **46.0** |

ViT-base architecture. We fine-tune the model for 12 epochs and evaluate the performance on the validation set. The results are summarized in Table 11. ExtreMA outperforms DINO/MSN/MoCo-v3/BEiT while using a lot less compute. ExtreMA outperforms MAE with the same number of pretraining epochs, but underforms MAE if MAE is trained longer. When pretrained on the ImageNet22k dataset, ExtreMA improves the performance by about 1% AP.

## 6 Conclusions

This work explores masking as a novel augmentation for Siamese representation learning. The investigated approach, ExtreMA, learns strong instance and distributed representations through data augmentations without masked modeling supervision. This work is inspired by the masking operation in masked modeling. However, it makes no claims that ExtreMA works in a similar way as masked modeling, since the learning objectives are different. ExtreMA exhibits several unique characteristics: 1) the use of extremely large masking ratios, 75%-90%; 2) fast convergence speed with multi-masking and scalability to large data; 3) low consumption of computational resources. Its ability on encoding precise locality for the instance representation may open up a new possibility for detection transfer.

## Broader Impacts

The proposed method learns representations from a specific dataset and as such may reflect biases contained in the dataset. Building a downstream application which finetunes the model on a customized dataset may also reflect bias and potentially negative societal impacts in the pretrained representations. The study in this paper is limited to curated datasets, and research on uncurated datasets warrants future research.

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

# A    Discussions on CaiT-Style Architecture

ExtreMA follows the CaiT-style transformer architecture Touvron et al. (2021b), where the class token is appended later in the attention blocks. We find that such design is critical for ExtreMA to stabilize learning, whereas the conventional ViT class token design failed to converge properly. Additionally, we also investigate a third option on using average pooling across tokens to aggregate the holistic representation. In Figure 5, we plot the training loss and the kNN classification accuracy for different class token designs. The ViT class token design leads to unstable optimization, and average pooling finds a representation shortcut. The CaiT-style architecture works as desired.

It remains as a limitation of this work to fully understand the training dynamics for the class token design. We hypothesize that the problem originates from the Siamese networks processing input sequences with very different lengths. This makes the learning of the class token representation harder, when it is processed throughout the network.

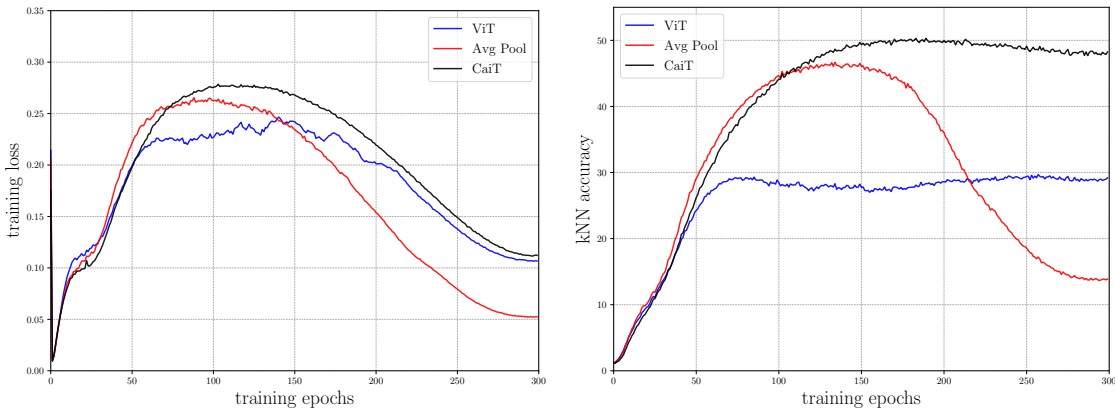

Figure 5: Training loss and kNN accuracy curves for three class token designs: ViT, average pooling, and CaiT. The network is trained with the ExtreMA objective with masking ratio 90% of 8 crops, using the ViT-small architecture.

# B    Multi-masking Performance Efficiency

ExtreMA greatly accelerates learning by multi-masking because the momentum encoder for processing the full tokens may be shared for multiple students. In the following Table 12, we further investigate its performance efficiency. We compare the model trained with 2 masks and 300 epochs against the model trained with a single mask and 600 epochs. The performance for these two settings are similar while multi-masking is significantly faster.

Table 12: Multi-masking performance efficiency.

| mask num. | epochs | ratio 75% | | | ratio 80% | | | ratio 90% | | |
|---|---|---|---|---|---|---|---|---|---|---|
| | | ft. | lin. | time | ft. | lin. | time | ft. | lin. | time |
| 1 | 300 | 82.4 | 64.8 | 30 hrs | 82.4 | 67.3 | 29 hrs | 82.3 | 61.6 | 28 hrs |
| 2 | 300 | 82.7 | 67.2 | 40 hrs | 82.6 | 68.8 | 36 hrs | 82.5 | 64.0 | 30 hrs |
| 1 | 600 | 82.6 | 67.8 | 60 hrs | 82.6 | 69.1 | 58 hrs | 82.5 | 63.4 | 58 hrs |

# C    Technical Details of the Generative Property

The pretrained ExtreMA model is able to inpaint the masked region given the visible patches. We follow the method of Deep Image Prior Ulyanov et al. (2018) to invert the representation back to pixels. Denote

the pretrained ViT encoder as $f(x)$ and an additional reconstruction network as $r_\theta(z_0)$, where $z_0$ is a fixed random noise and $\theta$ is the network parameter. Given an input image $x$ processed by a random mask $M$, the ViT encoder $f(M \cdot x)$ extracts feature representations for the visible tokens. The reconstruction network $r_\theta(z_0)$ generates a full image $x_0$, which is then passed to the encoder $f(x_0)$. We minimize a L2 energy function on the visible token representations between the masked input image and the reconstruction,

$$\min_\theta E\left(f(M \cdot x), M \cdot f(x_0)\right), \quad x_0 = r_\theta(z_0). \tag{1}$$

The optimization takes 3000 iterations with an Adam optimizer of learning rate 0.001. The pretrained ViT encoder remains fixed and the reconstruction network is optimized per input image. $x_0$ is the inpainted image.

## D    Additional Comparisons of Locality Properties

We compare the performance on localization with other works, MAE / DINO / MoCo-v3. We use the [cls] token representation from these models. In Figure 6, we find that DINO performs favorably well, and that MAE / MoCo-v3 degrades the performance notably. MAE does not supervise an instance representation in the formulation, and hence its instance representation is weaker. MoCo-v3 suffers from the heavy use of spatial cropping augmentation, and DINO improves by using small local crops for localization.

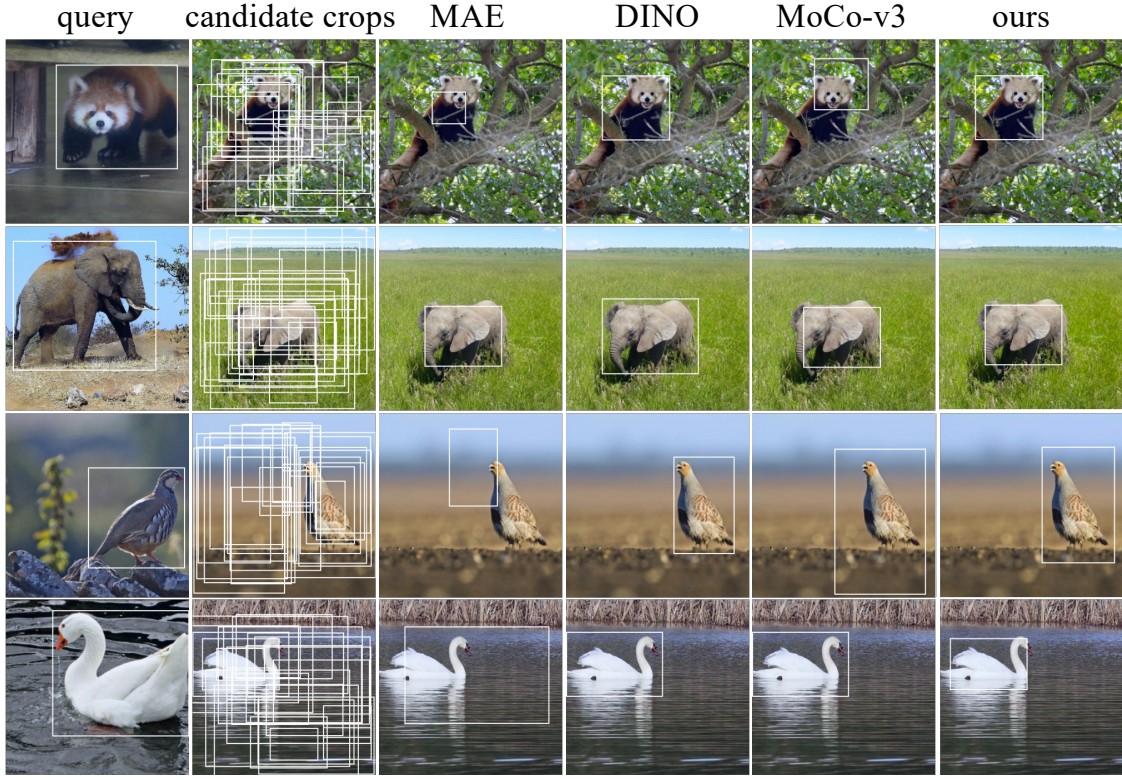

Figure 6: State-of-the-art comparisons with other models for localization.

## E    Details of Evaluation Protocols

The evaluation protocols for end-to-end finetuning and linear probing largely follow BEiT and MAE. The hyper-parameter configurations are detailed in Table 13 and Table 14. We finetune ViT-Small models for 200 epochs and ViT-Base models for 100 epochs. We use a base learning rate 1e-3 and layer decay 0.75 for

ImageNet1k pretrained models, and a slightly smaller learning rate 5e-4 and a smaller layer decay 0.65 for ImageNet22k pretrained models. The linear probing configuration is adopted consistently for all reported entries.

Table 13: End-to-end fine-tuning protocol.

| config | value |
|---|---|
| optimizer | AdamW |
| base learning rate | 1e-3 |
| weight decay | 0.05 |
| optimizer momentum | $\beta_1, \beta_2{=}0.9, 0.999$ |
| layer-wise lr decay | 0.75 |
| batch size | 1024 |
| learning rate schedule | cosine decay |
| warmup epochs | 5 |
| training epochs | 200 (S), 100 (B) |
| augmentation | RandAug (9, 0.5) |
| label smoothing | 0.1 |
| mixup | 0.8 |
| cutmix | 1.0 |
| drop path | 0.1 |

Table 14: Linear probing protocol.

| config | value |
|---|---|
| optimizer | LARS |
| base learning rate | 0.1 |
| weight decay | 0 |
| optimizer momentum | 0.9 |
| batch size | 4096 |
| learning rate schedule | cosine decay |
| warmup epochs | 10 |
| training epochs | 90 |
| augmentation | RandomResizedCrop |

## F    Additional Visualizations

We provide additional visualizations on the generative aspects of our model in Fig. 7, and attention maps of the distributed representations in Fig. 9. Both visualizations reveal properties of the distributed representations. These representations maintain accurate correspondences with the input tokens, while inferring meaningful semantic relationships among tokens. We also append failure examples on the reconstructions in Figure 8. The feature inversion technique fails to inpaint novel semantic areas especially when the masking ratio is large.

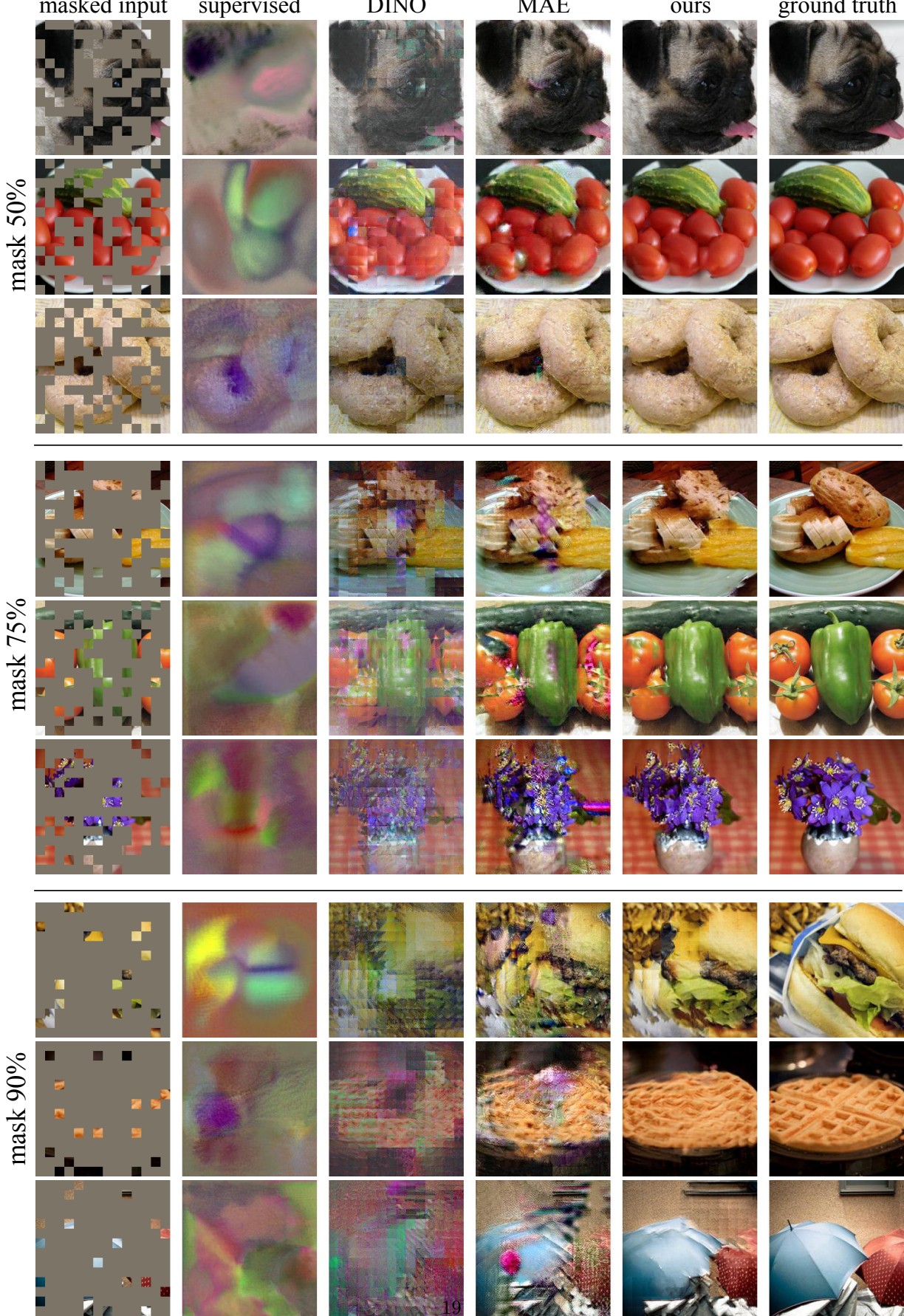

Figure 7: **Additional examples of inpainting at various masking ratios.**

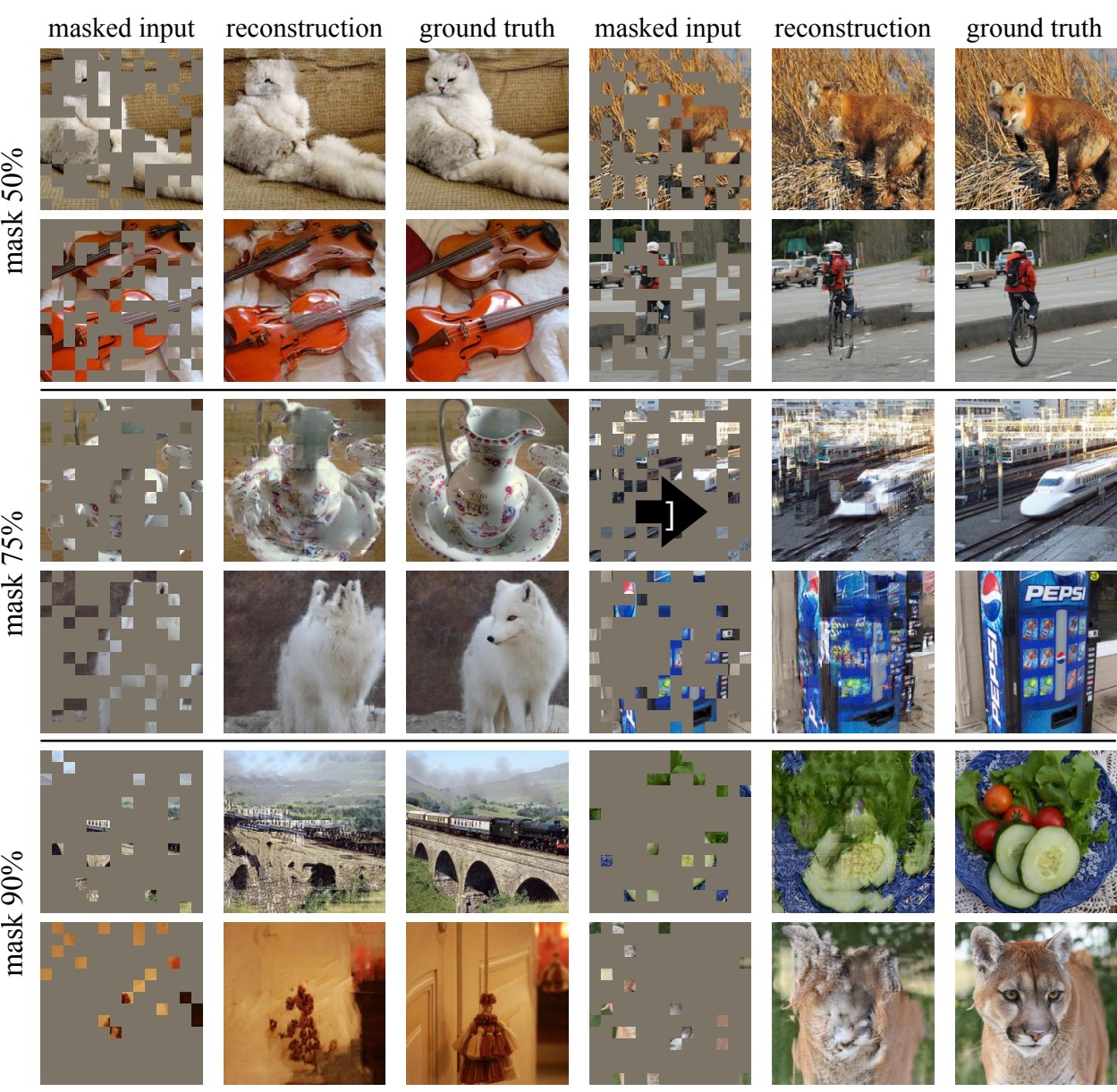

Figure 8: **Failure examples of our reconstructions on the generative aspect of our model.**

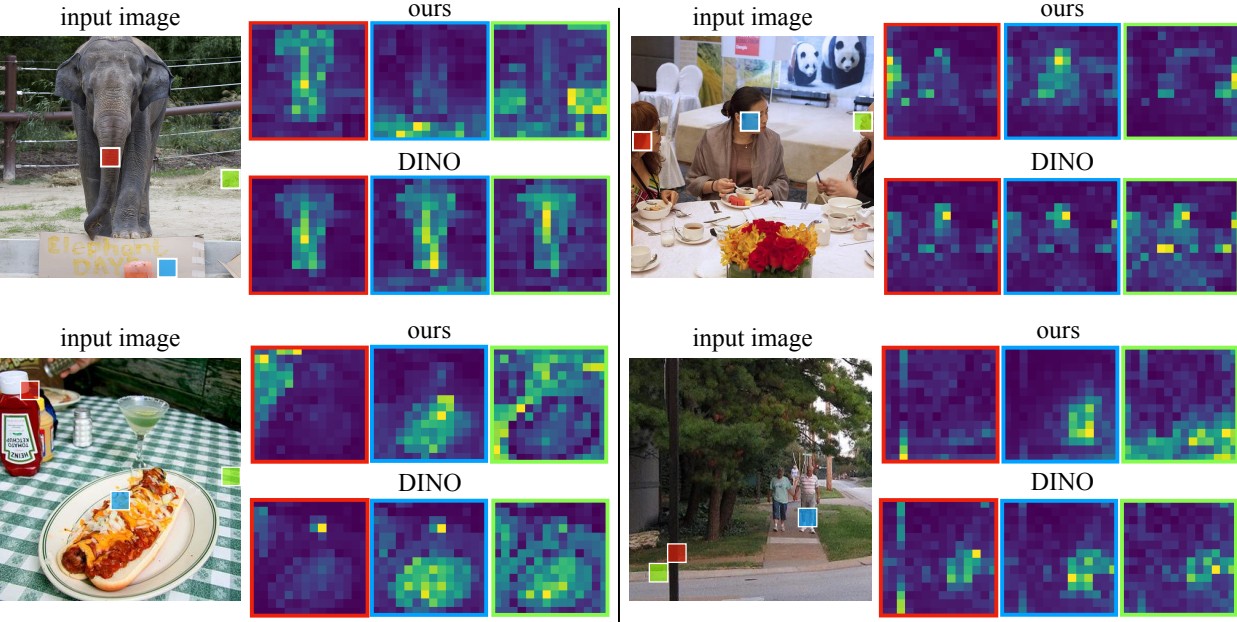

Figure 9: **Attention maps on the last layer of the ViT encoder.** We average the responses for 12 attention heads for visualization. Our model produces diverse and distributed attention maps, whereas DINO Caron et al. (2020) mainly attends to the foreground object, ignoring the others. The border color of the attention map corresponds to the colored query in the input image.

