# OpenReview forum: "Extreme Masking for Learning Instance and Distributed Visual Representations"
_TMLR — Accepted by TMLR_

### Review · Reviewer_h3DV · 2022-12-12

**Summary Of Contributions:**

This paper proposes a novel method for self-supervised representation learning for downstream recognition tasks. The key idea is two fold:

(a) Use a cross attention layer on top of a Vision Transformer backbone to obtain image level representations (as opposed to global average pooling).

(b) Use instance discrimination as the self-supervision proxy task but use "extreme" masking as an aggressive data augmentation strategy. Extreme masking here refers to the kind of masking done in Masked Autoencoders but with an even higher masking percentage (~80%)

The findings are:

(a) The proposed method benefits from extreme masking up-to even 90%.

(b) It learns representations that are both suitable for image level tasks (image classification) and spatial tasks (object detection, semantic segmentation, instance segmentation)

(c) The method benefits from pre-training on bigger datasets (ImNet-22k > ImNet-1k).

(d) Linear probing performance of MAE based methods is now on-par with SoTA contrastive methods (previously these were much worse, Table 7).


--------
### Post discussion update

My concerns have been addressed. I have updated "claims and evidence" to "yes".

**Audience:**

Yes

**Broader Impact Concerns:**

The paper does not include a broader impact statement. The proposed method does not directly create ethical concerns, but all recognition methods especially those that do object detection could have ethical implications. A brief comment about this would be great.

**Claims And Evidence:**

Yes

**Requested Changes:**

### Minor changes
--------------------------------------
In page 4: "or efficient for masked modeling ..."  ... I think you mean that the decoder needs to be rerun for each mask, making it inefficient. Or did I misunderstand this. Please clarify in text.

In section 5.2.: You could revisit, in the first paragraph, the point made in the introduction about MAE methods not being suitable for a direct evaluation.

---------------------------------
### Major changes
---------------------------------
Addressing weaknesses (1), (3) would be important for acceptance. Addressing these would flip by answer under "Claims and evidence" to Yes. (2) would be great and the "bonus" would strengthen the paper.

Please include a discussion of the limitations of the method (could be in the appendix as well).


**Strengths And Weaknesses:**

### Strenghts
------------------------
1. The paper identifies a gap between MAE based methods and contrastive methods for downstream instance level tasks and the need for both image level and spatially discriminative representations. The proposed method serves to both bridge this gap and provide a versatile representation that can be used for various downstream tasks.

2. The paper is well written. The analysis of the representation in figures 2 and 3 is particularly helpful.

3. Extensive ablations on masking ratio, multi-masking (using more than 1 mask for the student network), and an analysis of why the performance drops with more masks in Table 2, help the reader understand the efficacy of the proposed method and its hyper-parameters.

4. The paper makes some interesting and creative choices:
(a) An asymmetric instance discrimination loss suitable for the proposed method,
(b) A novel shared crop + color augmentation strategy that helps get high linear eval and fine-tuning Top-1 accuracy on ImNet-1k.

------------------------

### Weaknesses
------------------------
1. The effect of multi-masking in studied by controlling the number of training iterations. However, in most use cases the limiting resource is computation time. Is multi-masking beneficial when trained for the same number of total hours?

2. Figure 3: Are these results cherry picked or purely random. Please include some failure cases.

3. No error bars are included in any of the results. This is likely due to the computational requirements of each experiment. One work around is to include error bars for the default method (Say ExtreMA (80% ratio x N) ) for downstream ImageNet-1k linear eval. This will help readers when they try to reproduce the results and also for understanding which comparisons might be within error bars.

Bonus: The model is trained only on ImNet-1k / ImNet-22k. The readers would benefit from knowing what happens when the model is trained on long tailed data such as random video frames or scene centric images such as in the MS-COCO dataset.

---

> ### Author Response · Authors · 2022-12-16
> **Rebuttal to Reviewer h3DV**
>
> We thank the reviewer for the detailed comments. Minor revision suggestions and contents regarding broader impacts and limitations have been updated and highlighted in red. In the following, we respond to individual questions.
>
> 1. Yes, multi-masking is beneficial when trained under the same wall time. The  ExtreMA model performance is affected by the total number of the masking operations throughout training iterations. For example, multi-masking with 2 masks trained for 300 epochs performs similarly to 1 mask trained for 600 epochs. Since multi-masking shares a single momentum encoder, this makes multi-masking more computationally efficient than training with a single mask. For example, using 5 masks at the ratio of 80\% only doubles the training time compared against using one mask in Table 8.
> 2. We thank the reviewer for this great suggestion. Failure cases are appended in Figure 8. The major pattern of failure cases is the inability to inpaint and hallucinate unseen semantic areas. This is likely due to the nature of the inversion technique, which only probes the encoder without a well-trained decoder.
> 3. We thank the reviewer for raising the concern. We observe that the model can be reproduced stably within 0.1 percent for both linear probing and finetuning. We ran the model for 3 runs under the 80\% x 5  setting, attaining the linear probing performance 73.26, 73.23 and 73.28  respectively. We kindly note that prior contrastive models e.g. DINO / BYOL also enjoy similar stable reproducibility. We hope this can alleviate the concern.
> 4. (bonus) Exploring self-supervised representation learning in the wild on long-tailed uncurated dataset is of great importance for addressing research questions and application needs. However, drawing sensible conclusions from uncurated data requires careful ablation designs. For example, MS-COCO is an order of smaller than ImageNet. The iNaturallist is larger but limited to species. This is an interested question and we will leave it for future work.

---

> > ### Comment · Reviewer_h3DV · 2022-12-16
> > **Unsupervised [cls] token in baselines**
> >
> > I thank the authors for addressing my concerns.
> >
> > 1. Could this comparison point (2x masks 300 epochs = 1x mask 600 epochs) be added to the paper, in case it is not already there.
> > 2. Thank you for including failure cases and a limitations section.
> > 3. Could the variance be reported in the appendix? Nice that this is so low.
> > 4. Acknowledged
> >
> > Regarding the [cls] token being unsupervised in BeiT and MAE, would it be better to report their linear eval performance under average pooling?

---

> > > ### Author Response · Authors · 2022-12-23
> > > **Rebuttal to additional questions**
> > >
> > > We thank the reviewer for the detailed suggestions. We have updated the submission with a comparison regarding multi-masking and a footnote regarding variance.  Please check and let us know if you have further revision suggestions.
> > >
> > > For the additional concern on the choice of average pooling and [cls] token in baselines, we obtain 66.8% linear evaluation performance for average pooling using the official MAE code. This is 1% lower than the [cls] token with 67.8% accuracy. BEiT only releases the pretrained model on the ImageNet22k data. A fair comparison on ImageNet-1k is not immediately possible. Since BEiT linear evaluation performance is substantially worse, we believe the additional result on MAE is sufficient.

---

> > > > ### Comment · Reviewer_h3DV · 2022-12-25
> > > > **Reply to rebuttal to additional questions**
> > > >
> > > > My concerns have been addressed. Thank you!

---

### Review · Reviewer_xj2Y · 2022-12-26

**Summary Of Contributions:**

The paper proposes a method for self-supervised representation learning from images that can be seen as a combination of masked modeling and BYOL - namely, masking (with large masked fractions, around 80%) is used as an augmentation in the BYOL training setting. Note that, unlike data2vec, training is performed on the aggregated whole-image representation, not on the spatially distributed per-token representation. The method is evaluated on a few tasks (ImageNet classification, semi-supervised classification, instance segmentation, semantic segmentation) and shown to perform favorably compared to baselines while being faster to (pre-)train.

**Audience:**

Yes

**Broader Impact Concerns:**

The paper works on improving self-supervised image representation learning, I don't see any specific concerns, other than that generally better computer vision systems can be used both for good and for bad.

**Claims And Evidence:**

No

**Requested Changes:**

Fix all the issues mentioned in "weaknesses" (see there for more detail):
1. (critical) Empirically compare to data2vec and add a somewhat detailed discussion of the relationship between data2vec and the proposed method
2. (very desirable) Add apples-to-apples comparisons of training on whole-image vs spatially distributed per-token representations
3. (critical) Rephrase unsupported claims or add evidence
4. (critical) Fix writing/presentation issues

**Strengths And Weaknesses:**

Strengths:

1. The idea of the method is reasonable and interesting
2. Good empirical results on a few vision tasks (ImageNet classification, semi-supervised classification, instance segmentation, semantic segmentation), while training is quite a bit faster than some competing methods.
3. Good ablation studies of different aspects of the method
4. Mostly quite good writing

Weaknesses:

1. No empirical comparisons to data2vec and a limited discussion of the relationship between data2vec and the proposed method, while as far as I can tell the approach is very related, the main conceptual difference being training on aggregated whole-image vs spatially distributed per-token representation (as well as many other technical details). There should be a proper discussion of the relationship and a proper comparison.

2. Related to the previous, there is no systematic evaluation of aggregated whole-image vs spatially distributed per-token representations, while training on whole-image representation is more or less the main contribution of the paper. It would be very helpful to do some apples-to-apples comparisons, in terms of training speed and final performance. Otherwise, it remains unclear if the good performance of the method is specifically thanks to the use of whole-image representation or some other tricks.

3. Some unfounded statements:
3a. "the self-supervision from conventional augmentations lead to a common representation vector that encompasses multiple augmentations of an instance, and this invariance degrades the sensitivity of the representation to spatial locality" - need to support this statement with a reference or experiment.
3b. "The ability of ExtreMA to succeed with extreme masking is in part due to the momentum encoder processing the entire image, whereas the encoder in the masked image modeling never receives the full view as input. This creates train-test discrepancy for masked modeling, as the attention blocks need to generalize from processing a fraction of the tokens to the full set." - is there direct experimental evidence to support this? Stated as a fact, although I believe this is an unconfirmed hypothesis.
3c. "Extreme masking provides two other critical benefits..." etc - this paragraph needs empirical support, I would advise the authors to mostly stick to describing the method in the method section and analyze its properties later in the experiments.


4. Writing/presentation issues:
4a. Many papers cited as arxiv, while they have been officially published already
4b. Inconsistent use of "vision transformer" and "visual transformer". ViT stands for "vision transformer".
4c. Terms "instance" and "holistic instance" are not quite clear, in my opinion. I think by these terms the authors mean the whole image here, but I would say a more standard meaning of "instance" would be one object in an image, like in "instance segmentation". I would recommend rephrasing these, or at least very clearly explaining what is meant.
4d. The use of the term "distributed representation" is also confusing - here the authors mean the "per-token", "spatially distributed" representation before pooling over the image, while a more standard meaning, as far as I know, has to do not with the spatial organization of features but with how features code for the properties of the model input (see e.g. http://www.cs.toronto.edu/~bonner/courses/2014s/csc321/lectures/lec5.pdf, slide 6; or https://web.stanford.edu/~jlmcc/papers/PDP/Chapter3.pdf)
4e. "The space of masking augmentations becomes combinatorially smaller when the masking ratio increases." I do not quite get this statement. Yes, the space of 90%-masking masks if smaller than the space of 70%-masking masks, but still it's a huge space - assuming 196 patches, C(196,19) is over 10^26, which is way-way more than the model will ever be able to observe during training. How does it explain overfitting? I personally would vaguely hypothesize that the model overfits because the task is very difficult and the model tends to learn images by heart as a shortcut.
4f. "This is different from “cheating”, as the feature effectively learns semantics on the training data." - I do not get this phrase, please rephrase. I guess the intended meaning is that the model does not fail on representation learning completely like some other self-supervised methods, but this needs to be expressed more clearly.
4g.  "The overfitting phenomenon simply suggests that extreme masking is hungry for more training data" - this sentence does not sound exactly scientific, better rephrase. Also, yes overfitting could be solved by using more data, and it could be also solved in other ways, why state specifically this one here?
4h. "The distributed representations are frozen without updates, and this makes the cross-attention blocks lightweight with O(N) complexity compared to O(N^2) in self-attention" - I don't get it, what does the complexity of cross-attention have to do with the representations being frozen? And why/when are they frozen?
4i. "Conventional BYOL adopts a symmetric loss where the two views are learned to predict each other. <...> Thus, we adopt the asymmetric loss." - I'm unsure what's meant by symmetric and asymmetric here, clarifications are needed. Would be good to add a formula for the loss too.
4j. "Specifically, we follow the technique of deep image prior (Ulyanov et al., 2018) and minimize the L2 distance on the visible representations between the masked image and the reconstruction." - this needs to be explained in much more detail (potentially in the appendix). I know the Deep Image Prior and still, it's far from obvious how exactly it's used. Some readers may not know it at all.
4k. Visualizations in Section 4 are interesting but purely qualitative, so it's difficult to really make any proper conclusions based on them.
4l. Why some results (say, tables 1-7) are with linear probing, while others with kNN (say, Figure 4)? Would be nice to report kNN in tables too.
4m. "our model wants more data" - sounds unscientific, better smth like "benefits from more data" or "can make use of more data"
4n. "Such hardware requirement is amenable to resource-limited academic labs" - I'm not sure the word "amenable" works here

---

> ### Author Response · Authors · 2023-01-03
> **Rebuttal to reviewer xj2Y**
>
> We thank the reviewer for the detailed and valuable feedbacks. The comments helped us improve the paper substantially. We updated the manuscript as suggested, with changes highlighted in red. We address individual concerns as follows.
>
> 1. On data2vec. We expanded the discussions of the relationship to data2vec in Sec 2 and included a comparison in Table 7 and Table 8. Data2vec follows masked modeling and ExtreMA follows contrastive representation learning. They differ not only in the type of the supervision (whole-image or per-token), but also importantly in whether a mask token is being modeled.  For empirical comparisons, data2vec additionally uses layer-averaged targets which are shown to improve the performance substantially. Thus this makes the comparison at a system-level and it does not faithfully ablate the difference between whole-image or per-token supervisions.
>
> 2. A main contribution of our work is to show random masking can be used as a strong data augmentation for siamese representation learning. This differentiates our work from prior literatures. Our work does not try to reveal and address the connection between masked modeling and contrastive representation learning. We believe that this is an important research problem and it worths a separate paper. Ablating whole-image and per-token supervisions essentially asks for understanding the difference between the two representation learning paradigms. A mask token needs to be modeled, either in the encoder (BEiT, Data2vec) or in the decoder (MAE), in order to formulate per-token supervisions.
>
> 3. Unfounded Statements.
> - 3a. Reference Tian et al., 2020 and Purushwalkam & Gupta, 2020 are added to support the statement.
> - 3b. The statement is weakened as a hypothesis, not as a fact.
> - 3c. The paragraph is revised to describe the multi-masking technique. It’s efficiency and its effect on various masking ratios is moved to the experimental section.
>
> 4. Writing issues.
> - Issues 4a/4b/4c/4f/4g/4j/4i/4m/4n are fixed.
> - 4d. In the recommended material [https://web.stanford.edu/~jlmcc/papers/PDP/Chapter3.pdf](https://web.stanford.edu/~jlmcc/papers/PDP/Chapter3.pdf), it describes the distributed representation as a representation where each entity is represented by a pattern of activity distributed over many computing elements. In our context, we intend to mean that an image is represented by a pattern spatially distributed over many patch representations. In this aspect, we think the term is not a big concern. We also checked seminal works related to distributed representations such as Mikolov 2013 and Mnih 2009, and we did not find other formal definitions for the term.
> - 4e. We notice the overfitting phenomenon when we evaluate the model using linear probing. The performance on the training set is higher than non-overfitted models while the performance on the validation set is lower. This shows that the model successfully learns meaningful semantical representations from the training data, but less generalizable.
> The space of masking at ratio 90% is indeed very high, with C(196, 19) over 10^26. However, the visual tokens are redundant with high correlations. This largely reduces the number of unique masked input. Additionally, though C(196, 19) seems to be big, it is still significantly smaller than C(196, 49) (about 10^46) at ratio 75%. Nonetheless, we agree with the reviewer that the reason for overfitting needs more justification. We have weakened the statement about the complexity of masking.
>
> - 4h. The detailed design of the cross-attention blocks faithfully follows the CaiT paper. The term ``frozen'' means that patch tokens are not used as queries in the attention block to update their representations. In CaiT, only the [cls] token is used as the query to attend to other tokens. Thus the complexity becomes 1 (number of queries) x N (number of keys). We have revised the paper to make this clearer.
> - 4l. Linear probing and finetuning are used as the evaluation protocol for image classification. kNN is not a popular choice for benchmarking self-supervised models. For example, masked modeling approaches e.g. BEiT/MAE do not report performance using kNN. In Figure 4, kNN is adopted to visualize the convergence property under training iterations, since it is a lot faster than optimizing a separate classifier on ImageNet.

---

> > ### Comment · Reviewer_xj2Y · 2023-02-01
> > **thanks and follow-up comments/requests**
> >
> > Thanks for the detailed response and it's good to see that the paper improved. However, I have some follow-up comments/requests, please see below.
> >
> > 1. data2vec. Thanks for the update, this is much better. However, still not great: the phrase "... and its formulation inherits masked modeling" is a bit unclear and also incomplete - while yes data2vec is related to masked modeling, it is also very related to BYOL since it predicts the features it learns itself (with momentum encoder). As long as I can tell, the only conceptual difference between the proposed method and data2vec is whether the representation that's being predicted is pooled or not, and I think it should be clearly stated. Please be more precise here.
> > 2. Direct comparison of per-token vs whole-image supervision. I understand that doing extra experiments takes extra work (tbh it feels like in this case it shouldn't be super difficult, but admittedly I don't perfectly know all the technical details and tuning everything well can indeed take a substantial amount of time and effort), but it would also make the paper stronger and less borderline. It might be ok to accept the paper without these if everything else is good, but generally it's in the interest of the authors to make the paper stronger to increase the potential future impact.
> > 3. Unfounded statements. Thanks, looks better!
> > 4. Writing issues
> > 4d. While it's tricky to find a precise "ground truth" definition, in practice "distributed" usually refers to distributed over the dimensions of the feature vector, not over space. So writing "spatially distributed" helps resolving this potential confusion. Thanks for improving it, I think could be made even clearer - for instance, in the first sentence of the abstract.
> > 4e. I wasn't arguing with the fact that there are fewer 90-% masks, the point was that it's still a huge number, the model basically never sees the same mask twice, so it's not very clear what is meant by "overfitting". It would be great if the authors tried to explain it better. The current phrase "We hypothesize that space of masking augmentation becomes combinatorially smaller" is weird - the space is clearly smaller, that's not a hypothesis - the hypothesis is that it leads to overfitting, and it's unclear to me where this hypothesis stems from given the above. Please clarify.
> > 4h. Thanks! Sounds good.
> > 4l. Thanks, makes sense. Please mention this in the paper too.

---

> > > ### Author Response · Authors · 2023-02-04
> > > **follow-up responses**
> > >
> > > We thank the reviewer for checking each item and raising follow-up questions. We respond to the concerns as follows.
> > >
> > > 1. data2vec. We wish to clarify that the conceptual difference to data2vec is not only about whether the representation is being pooled or not. In data2vec, the mask tokens are used to model the invisible patches, and the supervisions are applied on these mask tokens instead of the visible tokens. This is the key idea of masked modeling. In ExtreMA, no mask tokens are used throughout the formulation and the supervisions are applied on the visible (instance) representations, instead of the masked representations. While there are apparent similarities between the two work, we believe the model formulations differ in important ways.
> > >
> > > 2. Following the prior response, replacing the whole image supervision with per-token supervision does not make ExtreMA to be data2vec, and vice versa.  Thus, it would be less of interest to conduct such comparisons. We would be glad to conduct this comparison if the reviewer finds it crucial.
> > >
> > > 4d. Thanks! The phrase is revised in the abstract.
> > >
> > > 4e. In conventional supervised learning, overfitting is observed by higher training accuracy and lower testing accuracy. In our self-supervised experiment, similar observations are found when a pretrained model is evaluated using linear probing. The pretrained representations have a higher linear probing accuracy on the training set but lower accuracy on the testing set. Overfitting is simply an observational fact.
> > > We understand that our hypothetical cause of the overfitting phenomenon may be misleading and unfounded. We thus remove the text about this hypothesis throughout the paper and stick to descriptions about the overfitting observations.

---

> > > > ### Comment · Reviewer_xj2Y · 2023-02-10
> > > > **re data2vec**
> > > >
> > > > Thanks for the responses and the further improvements.
> > > >
> > > > Re data2vec - I'm still not convinced by the argument about predicting only masked tokens. data2vec could also predict all tokens, not just masked - predicting the unmasked ones is trivial, but could be done and I'm pretty sure it wouldn't change basically anything. And then again the only major difference to the proposed method seems to be not pooling vs pooling.
> > > >
> > > > To be clear, I think it's fine if the proposed method only differs from data2vec in this way, it can still be interesting. But what is critical is to be clear and upfront about relation to prior methods and at the moment it still seems to me the discussion of the relation to data2vec is incomplete and obscure.

---

> > > > > ### Author Response · Authors · 2023-02-11
> > > > > **difference to data2vec**
> > > > >
> > > > > We respectfully disagree with the reviewer. Please allow us to defend this once more.
> > > > >
> > > > > All masked image modeling approaches use the [mask] token as input either to the encoder (BEiT, data2vec) or to the decoder (MAE). On the contrary, ExtreMA does not use the [mask] token. Whether the model processes the [mask] token or not is also one notable difference between data2vec and ExtreMA.
> > > > >
> > > > > For ExtreMA, it is possible that we replace the pooled supervision by per-token supervision at each visible patches. However, this alternative model is not identical to data2vec. Thus, whether the supervision being pooled or not is not the only difference between our work and data2vec.

---

> > > > > > ### Comment · Reviewer_xj2Y · 2023-02-13
> > > > > > **add more detailed discussion**
> > > > > >
> > > > > > I'm not saying the method is identical to data2vec, I'm saying it is related enough to warrant a detailed discussion in the paper, in particular for instance the argument about mask token as presented above.
> > > > > >
> > > > > > Btw I must say I don't exactly get the argument - [mask] tokens, especially if only added in the decoder, don't seem like such a bit deal, more of a technicality. And I'm curious if predicting the features of just the non-masked tokens BYOL-style would work at all - perhaps not, but who knows.

---

> > > > > > > ### Author Response · Authors · 2023-02-16
> > > > > > > **detailed discussions to data2vec**
> > > > > > >
> > > > > > > We thank the reviewer for the comment. Now, we are certain that there is no misunderstandings. We revised the discussion to data2vec as follows:
> > > > > > >
> > > > > > > ``` Data2vec formulates a masked image prediction task, with the prediction target bootstrapped from a momentum encoder. Similar to Data2vec, our work explores representation learning with the Siamese architecture and the masking mechanism. Differently, our model predicts the holistic instance representation without using the mask token throughout the network. ```
> > > > > > >
> > > > > > > We hope that the concern can be resolved. Thanks again for your efforts in improving the quality of our work!

---

> ### Author Response · Authors · 2023-01-23
> **Note on the rebuttal to reviewer xj2Y**
>
> Dear reviewer xj2Y,
>
> We submitted the rebuttal earlier, but the chosen visibility might not include everyone. We have fixed it and we hope that you are able to read the rebuttal. We apologize for the inconvenience.
>
> Thanks,
> Authors

---

> > ### Comment · Reviewer_xj2Y · 2023-01-29
> > **thanks, will responds soon**
> >
> > Hello,
> >
> > Indeed I didn't see the comment initially, saw it now (with a few days of delay, my bad), will respond within a couple of days.
> >
> > Best,
> > Reviewer

---

### Review · Reviewer_g5tf · 2023-01-16

**Summary Of Contributions:**

Self-supervised learning of visual representations without human annotations now achieves accuracies comparable to those of supervised learning, but research continues to seek higher accuracies, more computational efficiency, and more generality w.r.t. input data and its augmentations.
This work studies self-supervised representation learning for vision by student-teacher/self-distillation training of siamese networks where the teacher takes unmasked images and the student takes masked images.
In general, this follows asymmetric siamese network training methods like MoCo, BYOL, DINO, and the like, but this work focuses more specifically on two aspects: (1) masking as the input transformation for self-distillation and (2) designing the attention architecture and learning targets for efficient training.
For masking, a high rate (75-90%) is applied to the student input, which is indeed higher than the rate for alternative methods.
This contributes an existence proof for the use of such high masking, in showing experimentally that it can improve over other masking approaches like the MAE, and in particular it achieves
higher linear probing accuracies than the standard MAE by its BYOL-like scheme (Table 7), but these are lower than other contrastive and self-distillation methods like MoCo and DINO.
For attention architecture, the vision transformer (ViT) for pre-training includes both its spatial or "distributed" tokens and its global or "instance" token (the CLS token), and the CLS token is selected as the representation for self-distillation.
That is, the CLS token of the teacher serves as the target for regression the CLS token of the student.
The choice of CLS token regression makes the method more efficient, compared to masking with the MAE, because it requires no decoding from its latents to the images.
This contributes a slightly different architecture for asymmetric siamese network learning.
Furthermore, the teacher representation of a single, unmasked input can be shared as the regression target of multiple student representations from multiple maskings of the same image.
This reduces computation, again relative to the MAE, and relative to alternative contrastive and self-distillation schemes that transform both inputs.
This contributes a more efficient training computation, both per step and in the total number of steps (Figure 4, Table 8).
The method as a whole is called ExtreMA, after its extreme masking, but it is likewise defined by its choice of attention architecture and learning targets.
The experiments contribute an ablation of ExtreMA's design choices and an evaluation of its self-supervised representation in to with multiple current methods that make use of masking (MAE, BEiT) and self-distillation (MoCo, DINO).
The ablation covers masking rate, multi-masking, additional data augmentations, architecture choices like cross-attention depth, and the choice of loss between BYOL (without negatives) and InfoNCE (with negatives).
The evaluation of the presentation covers ImageNet itself (the pre-training data) and downstream tasks (semantic segmentation on ADE20K, detection on COCO).

**Audience:**

Yes

**Broader Impact Concerns:**

This submission is about a general method for self-supervision of deep networks on image data and so it does not have any specific ethical concerns.
A broader impacts statement is included to warn against the possibility of bias in pre-training data and the need to check for issues when fine-tuning to a given downstream task.
While generic, this warning is sufficient for this work.



**Claims And Evidence:**

Yes

**Requested Changes:**

**Critical Changes for Acceptance**

- Please discuss the potentially related works [A, B], or explain in the response how they are not sufficiently related.
- Report the results for MSN as prior or concurrent work in the tables comparing with other methods on ImageNet and downstream tasks (Tables 7-11).
  It is likewise a masking and self-distillation method and so it deserves to be in these comparison.
- Specify how multiple masking interacts with training batches and updates.
  Are the epochs the number of passes through the dataset, or the number of updates made?
  This question is relevant because of multiple masking.
  For example, if there are two masks per input, then are twice the number of update steps made in an epoch?
  Or does optimization accumulate gradients across computational batches so that the same number of updates are made with or without multiple masking?
- Please clarify the architecture used for section 5.3 and whether or not the results for the comparison methods (1) control for the architecture and (2) are taken from the prior work or re-evaluated.
  This is needed because some results seem to differ from those reported elsewhere, such as the DINO results in this submission's Table 10 vs. Table 2 in MSN.
- Please clarify the linear probing evaluation of the compared methods, such as the MAE and data2vec, and whether or not the approach used in this evaluation differs from the original papers.

**Minor Changes for Improvement**

- Consider rephrasing Claim #2 about data hunger as more precise and measurable claim, perhaps about accuracy vs. the number of training inputs, or simply saying that ExtreMA continues to improve on larger datasets like IN-22k.
- Table 8: consider equalizing the number of epochs so that methods can be compared on the same scale. It is possible to see that there is a difference as it is, but one has to divide the number of epochs and hours for a closer comparison.

**Update following Rebuttal and Revision**

The critical changes were addressed by the rebuttal and corrected in the revision. While potential improvements remain the claims and the evidence are now in agreement with accurate, convincing, and clear materials.

I still hold that certain choices in the exposition are suboptimal for clarity and accessibility, such as the wording for "instance" and "distributed" and the discussion of "cheating" during training, but these have been discussed in this review and the review by xj2Y so the feedback is there for the authors to decide on. These are not issues that preclude acceptance.

**Strengths And Weaknesses:**

**Strengths**

- This work notes that MAEs train purely on masked inputs, and never observe _unmasked_ inputs during training, although all test inputs are unmasked.
  By regressing the representation of unmasked inputs from the masked inputs, this work is able to learn from images like test inputs.
  This is a correct and useful observation; however, it has been made in concurrent work like MSN (Assran et al. '22).
  Furthermore, the claim in this work that the teacher input avoids train-test discrepancy for the method is not entirely justified, because it is only the teacher that processes full images, and not the student.
- ExtreMA shows that extremely high masking ratios (75-90%) can be effective with the right asymmetric training scheme, which is in contrast to prior methods that used masking rates of around 60% or even as low as 30%.
- The proposed multiple maskings of the input are efficient and improve optimization. Multiple maskings are computationally efficient, in that the same encoding of the full input can be used as a target for each, and the proposed method has no decoder, which requires more computation in the original MAE. This reduces the per-step and total computation time (Table 8). Additionally, multiple maskings speed up convergence, and while they lead to overfitting on IN-1k, they further improves results for the larger dataset of IN-22k (while single masking does not improve over IN-1k for the same number of optimization steps).
- The proposed extreme masking and multi-masking are effective at the stated purpose of improving the accuracy of linear probing (Tables 1-3). However, there is little to no effect on fine-tuning accuracy with differences of less than 1 percentage point.

**Weaknesses**

- Key related work on masking and contrastive learning is not reflected in the claims nor included in related work.
  - ADIOS [A] adversarially optimizes input masks as a kind of data augmentation for contrastive learning and compares with random masking (at the same rate as the original MAE).
  This work is related by its consideration of masking for contrastive learning, although not to the rate of ExtreMA, and in its consideration of local/non-spurious representations (ADIOS, Table 5) without per-token supervision.
  - CAN [B] combines masking and contrasting, along with noise prediction, in order to learn representations that are transferrable by fine-tuning and linear probing, with better linear probing accuracy than the original MAE. While different in technique, CAN shares some goals with this work, such as improving linear probing accuracy above pure masking methods.
  - data2vec and MSN [Assran et al. ECCV'22] are credited in the related work, but not accounted for in the claimed contributions of this submission, in particular (1): data2vec, MSN, and the original MAE for that matter all apply masking for the self-supversion of attention networks.
- The method is not fully explained and self-contained.
  First, the architecture is not fully specified. The schematic of the architecure (Figure 1) and does not include the projection and prediction heads. While these are common to asymmetric siamese network methods, it is still preferable to identify their use in the student and teacher.
  Second, the use of multiple maskings of each input implies several design choices: how to alter the batch size (by growing it or having fewer inputs with more maskings), how to count steps for optimization (does this work measure an epoch as equal to one pass over the dataset, which would take more steps due to multi-masking, or as equal to the original number of batches without duplicates), and relatedly how to adjust hyperparameters accordingly.
- The use of multiple maskings per input is novel, but more narrowly, because multiple augmentations have been studied in prior work [C].
  More specifically, multiple augmentations for a single input during training has been applied for supervised learning [C].
  However, the prior work did not look at self-supervision, and it may include conflicting findings, in that it reports slower convergence on the training set but higher accuracy on the test set.
- Experiments are incomplete w.r.t. prior work and lack some controls.
  - It is not clear if the alternative representations are transferred and evaluated as intended in their original papers. For instance, MAE and adata2vec apply mean pooling to learned tokens for the purpose of classification (see the MAE code https://github.com/facebookresearch/mae or data2vec paper Sec. 4.1). However, this work comments that "Since no explicit supervisions are applied on the [cls] token in masked modeling, BEiT, MAE and Data2vec perform less competitive for linear probing." which suggests a different choice of input is used for linear probing.
  - Semi-supervised classification (Table 10) excludes MSN, which is cited in related work, but MSN reports comparable or better results for both ViT-B and ViT-S in this setting. Some reuslts of DINO are also higher in the paper than in this submission. Where are the differences?
  - The quality of reconstructions/inversions is only evaluated with qualitative examples (Figure 2), and it is not specified how these examples are chosen. Is this a random sampling or a cherry-picking of results? For what it's worth the column of ExtreMA results does look better, but there is no non-qualitative assessment, such as an FID score, or retrieval scores for matching inversions with the original input images across the dataset.
  - Localization (see "Locality Properties") is only evaluated with qualitative examples (Figure 3), but other work considers linear or nearest neighbor evaluations of local tasks like segmentation, such as DINO, which include ground truths and quantitative scores. As a first step, this work could at least evaluate the localization task of ILSVRC (assuming Figure 3 is showing images from ImageNet, which is not specified in the caption).
  - Multi-masking is proposed as a key part of ExtreMA, but multiple augmentations of the same input could likewise be applied in the other self-distillation methods. This may have some potential benefit for optimization with other types of augmentations too, as suggested by [C], and so it would be informative to compare with at least one of MoCo/DINO/etc. with multiple augmentations of each input.
- (Minor) The chosen terminology for "instance" and "distributed" representations is likely to confuse different kinds of readers.
  For example, "instance" could refer to a single input image, an object within an image, or a unique individual object across images (such as a particular person).
  At the same time, a "distributed" representation already has a defined meaning in neuroscience and machine learning that indicates a representation includes multiple dimensions or units.
  Consider terms that are shared by other work, such as image/global/holistic representations of a whole input image, and local/dense/spatial representations of the tokens across space.
  Choosing clearer terminology could work better than trying to fix the confusion with footnote 1.

**Summary**

This is a borderline submission. On the positive side, it does explore a different use of masking for visual representation learning, and highlights the use of extreme and multiple transformations of single inputs, and the experiments and ablations show improvements from these choices. This is either in contrast to other works on self-supervision, or in addition to findings for supervised learning (concerning the use of multiple augmentations of the same input).
On the negative side, the claimed contributions 1 & 3 are not entirely compatible with existing papers, and there are some concerns about the comparability of exprimental settings across papers.

- Claim #1. data2vec, MSN, and CAN (plus even the original MAE) could be considered to satisfy this claim. This should be sharpened into a claim about ExtreMA itself.
- Claim #2. This is explained in the paper and supported by clear exprimental evidence: multi-masking is computationally efficient, speeds up convergence, and improves accuracy given a larger pre-training dataset.
- Claim #3. Is this not already shown by works on self-supervised learning of image-level representations, like MoCo and DINO, which evaluate on global and local downstream tasks? Their level of supervision is also on the image or "instance" level of representation. This claim should be adjusted to highlight what further information is contributed by the downstream experiments for ExtreMA.

Note: I have marked "No" to the question on Claims and Evidence for now due to the discprepancies between the claims, results, and prior work and due to the lack of precision in some of the statements in the paper. However, I will reconsider this pending the response and revision provided for this submission.

**Other Feedback and Questions**

- "Learning requires the model to capture informative variations in an instance, instead of encouraging invariance." How? Invariance to transformations will also improve the accuracy of the regression for self-distillation. Is this saying that this learning target does not collapse, or in other words cannot be invariant to the input? If so, please re-phrase, perhaps by saying "instead of encouraging invariance (collapse) across all inputs".
- How is the overfitting of extreme masking on the training set, as discussed on page 4, different from "cheating"? If it "effectively learns semantics" than why does it not generalize. Please make this statement more precise.
- "CaiT-style architecture" likely indicates cross-attention transformers, as in Touvron et al. 2021b, but this abbreviation is not introduced or explained. Please spell out its first usage.
- Masked Contrastive Representation Learning [D] is yet another hybrid scheme for masking and contrasting, and while it is only on arxiv, and differs from ExtreMA in including a local reconstruction loss, it may nevertheless be worth a look. (To be clear: this arXiv has not been considered as prior work for the purpose of this review.)

**References**

- [A] [Adversarial masking for self-supervised learning](https://arxiv.org/abs/2201.13100). Shi et al. ICML'22.
- [B] [A simple, efficient and scalable contrastive masked autoencoder for learning visual representations](https://arxiv.org/abs/2210.16870). Mishra & Robinson et al. arXiv'22.
- [C] [Drawing Multiple Augmentation Samples Per Image During Training Efficiently Decreases Test Error](https://arxiv.org/abs/2105.13343). Fort et al. arXiv'21.
- [D] [Masked Contrastive Representation Learning](https://arxiv.org/abs/2211.06012). Yao et al. arXiv'22.

---

> ### Author Response · Authors · 2023-01-23
> **Rebuttal to reviewer g5tf, part 1**
>
> We would like to express our sincere gratitude for your time and efforts in reviewing our paper. We have updated the paper according your suggestions. In the following, we respond to your concerns and questions.
>
> Critical Concerns:
>
> 1. On paper novelty and related works.
>     - We hope to first clarify the novelty of our work. The key contribution is to show masking can be viewed as a strong data augmentation for contrastive representation learning. None of the prior works are able to make this claim. They either apply masking in conjunction with conventional data augmentations, or formulate a hybrid model of masked modeling and contrastive learning. Given the context that masked modeling (BeiT/MAE/Data2vec) emerges as a competitive approach for representation learning, we believe that showing contrastive representation learning can also be driven by masking is an important contribution.
>     - MSN also explored in this direction. However, masking is mainly useful for computation efficiency, not for the key learning signal. For example, in MSN’s Table 9, the model does not work with masking alone. It heavily relies on conventional image augmentations to work.
>     - We missed the ADIOS paper and we have included it as a related work. ADIOS applies masking in addition to conventional image augmentations on contrastive learning frameworks. It does not show masking itself could be strong.
>     - We missed the CAN paper and we have included it as a related work. We kindly note that CAN is on arXiv at late October and it is not officially published. CAN is a hybrid model of masked modeling and contrastive learning. It does not show masking could be a strong data augmentation for contrastive learning.
>     - The role of masking in Data2vec/MAE and ExtreMA are related but different. Both use random mask sampling to process the input tokens. They differ in the learning objective. Data2vec/MAE define the prediction target by masked tokens. ExtreMA’s prediction target no longer involves masking.
> 2. Multi-masking design choices. Yes, the same number of updates are made with or without multiple masking. In our implementation, the gradients for multiple maskings are averaged and then a single step of gradient step is taken. The learning rate scales with the batch size (the unique number of images) but not with the number of masks. This shows that the fast convergence property of multi-masking is a result of accurate gradient estimate, not larger optimization step size or more optimization updates. We have updated the texts to make this clear.
> 3. Feature representations for linear probing MAE/data2vec. We apologize for this misunderstanding. We actually follow prior works for using mean pooling to evaluate masked modeling approaches.  Please note that the numbers are consistent with the MAE paper. We have revised the quoted sentence to avoid the confusion.
> 4. The semi-supervised setting in the MSN paper. There are notable differences between our evaluation protocol and the one proposed in MSN. First, we fine-tune the full network end-to-end following SimCLR-v2, while MSN freezes the pretrained model and only tunes a single linear layer. Tuning a single layer may reduce overfitting, and thus it obtains better performance when labels are extremely scarce. For this reason, MSN focuses on 1,2 and 5 labeled samples per class.  Second, the control for the architecture is also different. We downloaded prior models with ViT-B/16 size and re-evaluated the performance in Table 10. MSN’s insight is to study performance under model scaling. The reported DINO performance is higher due to the use of larger architecture ViT-B/8, which is heavier than ViT-B/16.
>
>     To make the comparison fair, we downloaded the MSN ViT-B/16 model and evaluated its performance under our protocol. The performance is 66.6% with 1% of labels and 76.8% with 10% of label, which is on par with ExtreMA, attaining 67.3% and 76.1% respectively.
>
> 5. We have added the entry MSN in Table 7-11. Since MSN paper does not report the official results for these benchmarks, we will need to evaluate its released model on these downstream datasets. We promise to update the results in the next few days before the discussion period deadline. Based on our explanation about the novelties, we believe the performance of MSN will not challenge the novelty of ExtreMA.
> 6. Claims of the paper. Claim #1: We narrow the scope of the claim to siamese representation learning. Claim #2 : Multiple masking speeds up convergence and improves the performance with more data.  Claim #3: Changes are made to reflect the downstream performance of ExtreMA.

---

> > ### Comment · Reviewer_g5tf · 2023-02-01
> > **Thank you for the response! This addresses the critical changes.**
> >
> > > 1. On paper novelty and related works. We hope to first clarify the novelty of our work. The key contribution is to show masking can be viewed as a strong data augmentation for contrastive representation learning.
> >
> > Thank you. This is a more precise scope for the contribution, which is supported by the experiments, and distinct from prior work. The discussion of the related methods is accurate, and ExtreMA does indeed differ in its sole reliance on masking and its purpose in masking for siamese representation learning.
> >
> > > We missed the ADIOS paper and we have included it as a related work.
> > > We missed the CAN paper and we have included it as a related work.
> >
> > Thank you for making the related work more comprehensive, and therefore more informative to the reader interested in self-supervision by masking/contrasting/both.
> >
> > > 2. Multi-masking design choices.
> >
> > Thank you for these key clarifications and for specifying these details in the text. These are necessary to understand the results, and now they are known.
> >
> > > 3. Feature representations for linear probing MAE/data2vec. We apologize for this misunderstanding. We actually follow prior works
> >
> > Thank you for correcting this misunderstanding and revising the text.
> >
> > > 4. The semi-supervised setting in the MSN paper.
> > > 5. We have added the entry MSN in Table 7-11
> >
> > Thank you for addressing MSN and incorporating its results into the paper for comparison.
> >
> > > 6. Claims of the paper.
> >
> > Thank you for adjusting these to respect prior work and more exactly describe this work. The revised claims serve to better orient the reader.
> >
> > The full response and revision address the critical changes, so I will update my review and the Claims and Evidence question accordingly.

---

> > > ### Author Response · Authors · 2023-02-04
> > > **Thank you for the review!**
> > >
> > > Dear reviewer g5tf,
> > >
> > > It is glad that our revision and responses resolve your concerns. We wish to thank you for reviewing our paper. Your suggestions improve the quality of the paper substantially.
> > >
> > > Best,
> > > Authors

---

> ### Author Response · Authors · 2023-01-23
> **Rebuttal to reviewer g5tf, part 2**
>
> Other questions and feedbacks:
>
> 1. Multiple augmentation for supervised learning [C]. In [C] Figure 2/3, it reports faster convergence with multiple augmentations when the unique number of images is fixed. In [C] Figure 4/5, convergence is slower when the unique number of images is decreased with multiple augmentations. Our finding actually align with [C] when the batch size is fixed. However, there are a number of differences. The optimal learning rate and the number of parameter updates are dependent on the number of augmentations in [C]. In our case, convergence comparisons are more apparent because the number of parameter updates and learning rates are fixed. We have included a discussion to this work in the related work.
> 2. Multiple augmentations on MoCo and DINO. In our opinion, these work already take advantages of multiple augmentations in their frameworks. For example, MoCo’s loss formulation is symmetric: a→b, b→a. Two augmentations are used per image. DINO uses 2 global crops and 8 local crops, with each crop augmented from the original image. However, they do not formally study the convergence property for multiple augmentations. More importantly, multiple augmentations are  computationally more appealing for random masking, as each augmented view is lightweight.
> 3. We thank the reviewer for the suggestion. We have updated Figure 1 with projection and prediction heads. Details about the head networks can be found at subsection “BYOL Details”.
> 4. Qualitative experiments in Section 4. We agree with the reviewer that these qualitative studies remain as weaknesses to the paper. However, evaluating the reconstructions requires non-trivial efforts which may go beyond the purpose of the paper. For example, the reconstructions are slow (3000 iteration steps) and we may further need to tune the hyper-parameters to obtain the best FID scores. More visual examples including failure cases are shown in the appendix.
>
>     Our localization task performs retrieval on the [cls] token from a number of bounding boxes. The task is essentially one-shot object detection. This is different from DINO’s segmentation which uses attention maps to derive segmentations. The visualization of locality property is mainly to show ExtreMA’s [cls] token representation contains spatial and scale information. Benchmarking one-shot object detection will require non-trivial efforts. We hope that such limitation will not be a critical concern for the paper.
>
> 5. About invariance to transformations. Conventional contrastive learning approaches e.g. SimCLR/MoCo/BYOL encourages invariance to data augmentations to improve the representation quality. Typical invariances include spatial and scale invariances. ExtreMA learns to extrapolate from the masked input, but not to find the common invariant subspace between the views. We have rephrased this according to your suggestion.
> 6. How is the overfitting of extreme masking on the training set different from "cheating"?  We observe that for the linear probe evaluation, the overfitted model has a high accuracy on the training set but low accuracy on the validation set. This shows that the model captures the semantics during pretraining, but it is not able to generalize.

---

> > ### Comment · Reviewer_g5tf · 2023-02-01
> > **Thank you for the response, pt. 2**
> >
> > > 1. Multiple augmentation for supervised learning
> >
> > Thank you for providing this discussion, and for mentioning prior work on multiple augmentations in the revision. I now appreciate how the findings align with [C], and as before I acknowledge that ExtreMA contributes to this understanding in its study of multiple augmentation by masking.
> >
> > > 2. Multiple augmentations on MoCo and DINO
> >
> > Ok. MoCo and DINO do in fact use multiple augmentations, but the intention was to highlight that this work could study multiple masking in other contrastive schemes. However, it is fine to leave that to future work.
> >
> > > 3. We have updated Figure 1 with projection and prediction heads
> >
> > Thank you, the new figure is more clear, and indicates the asymmetric learning scheme too (with the predictor from the teacher view).
> >
> > > 4. Qualitative experiments in Section 4.
> >
> > I understand the difficulty of making these experiments quantitative. Including qualitative examples to cover both successes and failures is acceptable, if not ideal, because there are other tests of the learned representation (such as the downstream evaluations on segmentation and detection).
> >
> > > 6. How is the overfitting of extreme masking on the training set different from "cheating"?
> >
> > I am still unsure what is meant by "cheating" or not, and the response has not made this more precise. This is a minor point though, however it is phrased, as the results do make clear that there is an issue with overfitting and a method is introduced to remedy it. That is the part that matters and that is clear.

---

> ### Author Response · Authors · 2023-01-28
> **Results for MSN in Table 7-11**
>
> Hi Reviewer,
>
> We downloaded the official model from the MSN repo and evaluated its downstream performance. The results are included in Table 7-11. We observed that MSN is weaker than DINO for classification, detection and segmentation, perhaps because it is specifically tuned for semi-supervised classification. We hope this will resolve your concerns.
>
> Best,
> Authors

---

> > ### Comment · Reviewer_g5tf · 2023-02-01
> > **Thank you for including the MSN results!**
> >
> > Reporting the comparison with MSN in these tables satisfies the requested change in the review:
> > > Report the results for MSN as prior or concurrent work in the tables comparing with other methods on ImageNet and downstream tasks (Tables 7-11)

---

### Decision · Action_Editors · 2023-03-05

**Recommendation:** Accept as is

**Comment:**

All three reviewers lean towards acceptance of the paper. The AE thanks the authors and reviewers for engaging in productive discussions that have improved the manuscript. The AE supports accepting the paper as is -- there have been substantial revisions to address reviewer concerns.

**Audience:**

Yes. The findings are of clear interest to many of TMLR's audience.

**Claims And Evidence:**

Yes. The claims of the paper are supported by accurate, convincing, and clear evidence as supported by all three reviewers. The paper been improved by a number of rounds of revision, especially driven by one expert reviewer.